# Towards Minimizing Feature Drift in Model Merging: Layer-wise Task Vector Fusion for Adaptive Knowledge Integration

**Wenju Sun**[1,2], **Qingyong Li**[1,3,*], **Wen Wang**[1,2], **Yang Liu**[1,2], **Yangliao Geng**[1,2,*], **Boyang Li**[4]

[1]Key Laboratory of Big Data & Artificial Intelligence in Transportation, Beijing, China
[2]School of Computer Science and Technology, Beijing Jiaotong University, Beijing, China
[3]Frontiers Science Center for Smart High-Speed Railway System,
Beijing Jiaotong University, Beijing, China
[4]College of Computing and Data Science, Nanyang Technological University, Singapore
`{sunwenju, liqy, wangwen, yliucit, gengyla}@bjtu.edu.cn`
`boyang.li@ntu.edu.sg`

## Abstract

Multi-task model merging aims to consolidate knowledge from multiple fine-tuned task-specific experts into a unified model while minimizing performance degradation. Existing methods primarily approach this by minimizing differences between task-specific experts and the unified model, either from a parameter-level or a task-loss perspective. However, parameter-level methods exhibit a significant performance gap compared to the upper bound, while task-loss approaches entail costly secondary training procedures. In contrast, we observe that performance degradation closely correlates with feature drift, i.e., differences in feature representations of the same sample caused by model merging. Motivated by this observation, we propose Layer-wise Optimal Task Vector Merging (LOT Merging), a technique that explicitly minimizes feature drift between task-specific experts and the unified model in a layer-by-layer manner. LOT Merging can be formulated as a convex quadratic optimization problem, enabling us to analytically derive closed-form solutions for the parameters of linear and normalization layers. Consequently, LOT Merging achieves efficient model consolidation through basic matrix operations. Extensive experiments across vision and vision-language benchmarks demonstrate that LOT Merging significantly outperforms baseline methods, achieving improvements of up to 4.4% (ViT-B/32) over state-of-the-art approaches. The source code is available at `https://github.com/SunWenJu123/model-merging`.

## 1 Introduction

Pretrained foundational models [24, 4] encapsulate rich, transferable knowledge, which has facilitated their widespread use for fine-tuning on downstream tasks, yielding superior performance. However, the predominant use of the pretraining-finetuning paradigm has resulted in a proliferation of fine-tuned models, substantially increasing storage and maintenance costs for deployment. This challenge has driven the development of model merging, an effective strategy that consolidates the knowledge from multiple fine-tuned models into a single model, thereby eliminating the need for costly retraining.

Existing model merging methods typically focus on minimizing parameter differences between task-specific models and the merged model, such as arithmetic averaging of model parameters [12, 20] or generating masks based on heuristic factors [38, 5, 34]. These approaches aim to identify important

---

*Correspondence to: Yangli-ao Geng <gengyla@bjtu.edu.cn> Qingyong Li <liqy@bjtu.edu.cn>

parameters across different task models, attempting to preserve these key parameters during the merging process through weighting or masking strategies. While these methods are simple and efficient, their performance still lags behind the upper bound. Other approaches incorporate loss functions to train merging weights [41] or use adapter modules [40] during the merging phase. These techniques achieve impressive performance but are computationally expensive due to the need for secondary training.

In contrast to the aforementioned methods, our study investigates the performance degradation of merged models from the perspective of feature drift, i.e., differences in feature representations of the same sample caused by model merging. As illustrated in Figure 1(a), we observe a strong correlation between feature drift (measured by cosine distance) and the performance degradation associated with merging. Furthermore, Figure 1(b) demonstrates that feature drift becomes more pronounced as network depth increases, as small perturbations in the initial layers are progressively amplified through the layers of the network.

These findings motivate our proposal of Layer-wise Optimal Task vector Merging (LOT Merging), a method that explicitly minimizes feature drift across layers. Specifically, LOT Merging uses squared error to measure the feature difference between the merged model and task-specific models, which can be formulated as a convex quadratic optimization problem. Solving this problem analytically yields closed-form solutions for parameters in both linear and normalization layers, enabling efficient model consolidation via basic matrix operations. Additionally, we provide an intuitive explanation of LOT Merging through theoretical analysis in two extreme cases, illustrating its ability to adaptively adjust the merging strategy based on task dependencies. In summary, this paper makes the following contributions:

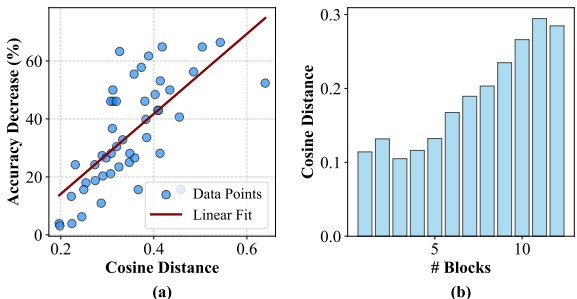

Figure 1: An illustration of feature drift (measured as the cosine distance between features extracted by task-specific expert models and the merged model) is presented using Task Arithmetic [12] on eight vision tasks. (a) The feature drift in the last layer shows a linear correlation with the accuracy decrease, with data points collected for each task under varying merging coefficients. (b) Feature drift becomes more pronounced as the network depth increases.

- We formulate the model merging task as a convex quadratic optimization problem by minimizing layer-wise feature drift, which enjoys a closed-form solution and such inspire an efficient model consolidation method referred to as LOT Merging.

- We provide a theoretical analysis of LOT Merging, revealing its resilient merging capabilities that effectively account for correlations among different task feature spaces.

- We conduct extensive experiments on a variety of visual and vision-language benchmarks, showing LOT Merging's superiority over state-of-the-art methods while maintaining robustness with limited exemplars.

## 2   Related Work

Model merging aims to integrate the knowledge of multiple fine-tuned models into a single model. Early work primarily focused on weighted averaging strategies, such as employing the Fisher information matrix [20], or minimizing the feature distance between the merged model and individual models [13]. Recent research on multi-task model merging is typically built upon "task vectors" [12], defined as the parameter differences between the pre-trained model and each fine-tuned counterpart. Task Arithmetic [12] demonstrates that simple operations on task vectors—such as addition—can be used to edit or merge knowledge effectively. Subsequent methods like Ties-Merging [38] and PCBMerging [5] improve upon this idea by pruning low-magnitude components from task vectors. Extensions to parameter-efficient fine-tuning settings include PEFT [42] and MoLE [35], which adapt Task Arithmetic to LoRA-based modules [10]. Additionally, some model merging methods rely on test-time training techniques. For instance, AdaMerging [41] learns a set of layer-wise merging coefficients via gradient descent; Surgery [40] introduces an auxiliary adapter module to

align intermediate representations; WEMoE [32] and Twin Merging [18] adopt mixture-of-experts frameworks and train a router to select among experts; and Localize-and-Stitch [7] constructs a binary mask that determines which parameters to merge. Since the training requires constructing computation graphs to achieve gradient descent, the memory and computation overhead of such methods tend to be expensive when applied to large-scale foundation models.

This paper introduces a layer-wise merging strategy for task vectors in parameter space, aiming to mitigate feature drift during model merging. By analyzing three parameter types in transformer architectures—linear weights, scaling factors, and bias terms—we derive closed-form solutions that enable efficient and principled integration. While our solution for linear weights parallels RegMean [13], our formulation operates on task vectors rather than original parameters, marking a fundamental distinction (see Section 5.2). Moreover, our method extends beyond RegMean by addressing additional parameter types and offering a deeper understanding of the mechanisms behind LOT merging. Compared to RegMean, our approach yields improvements of 10.9% (ViT-B/32) and 6.8% (ViT-L/14), and remains effective in dataless scenarios. Additionally, our method shares the goal of mitigating feature drift with Surgery [40] and CAT Merging [29], but differs in approach. Surgery introduces a test-time alignment module, whereas our method is entirely training-free. Unlike CAT Merging, which constructs bases or masks to trim task vectors, we derive a direct closed-form solution for optimal merging, resulting in further performance improvements.

## 3 Preliminary

### 3.1 Model Merging

**Problem setup.** Consider a pre-trained model characterized as an $L$-layer neural network $W_{\text{pre}} = \{W_{\text{pre}}^l\}_{l=1}^L$, where $W_{\text{pre}}^l$ represents the parameters of the $l$-th layer. Starting from $W_{\text{pre}}$, we fine-tune it independently on $K$ down-stream tasks, resulting in task-specific expert models $\{W_1, \ldots, W_K\}$. The objective is to merge these fine-tuned experts into a single unified model $W_{\text{mtl}}$ without the need for redundant retraining.

**Task Vector.** To facilitate model merging, Task Arithmetic [12] introduces the concept of *task vector*, defined as the parameter difference between the fine-tuned expert $W_k$ and $W_{\text{pre}}$:

$$T_k = W_k - W_{\text{pre}}, \tag{1}$$

where the arithmetic operations over parameter sets are applied layer-by-layer, i.e., $W_k - W_{\text{pre}} = \{W_k^l - W_{\text{pre}}^l\}_{l=1}^L$. Task Arithmetic has demonstrated that performing simple arithmetic operations over task vectors can effectively integrate knowledge into the pre-trained model:

$$W_{\text{mtl}}^{\text{ta}} = W_{\text{pre}} + \lambda \sum_{k=1}^K T_k, \tag{2}$$

where the addition is performed layer-wise and $\lambda$ is a manually defined scaling factor.

**Resource limitations.** In practice, it is often infeasible to access the full downstream training sets, yet most existing merging techniques still depend on some form of data-driven calibration. For instance, Fisher Merging [20], TATR [30], and CAT Merging [29] require a subset of labeled examples to estimate parameter importance, while approaches such as Task Arithmetic and Ties-Merging rely on a validation set to select hyperparameters. In addition, test-time adaptation methods require access to unlabeled examples at inference time to adjust the merging weights or train additional modules. These methods typically require iterative backpropagation, making them impractical under constrained GPU resources to merge large-scale models. In contrast, our method is entirely training-free, incurs only a handful of forward passes with a small sample set.

### 3.2 Negative Transfer

One major challenge in model merging is the negative transfer [38, 30], which occurs when the knowledge acquired by individual models conflicts or interferes with one another. Mathematically, negative transfer can be quantified by the degradation in performance across tasks resulting from the merging process. Let $\mathcal{L}_k(\cdot)$ denote the loss for task $k$, the negative transfer introduced by the merging

task vector $T$ (where for task arithmetic, $T = \lambda \sum_{k=1}^{K} T_k$) can then be defined as follows:

$$\Delta \mathcal{L}_k = \mathcal{L}_k \left( W_{\text{pre}} + T \right) - \mathcal{L}_k \left( W_k \right). \tag{3}$$

However, analyzing this degradation is often challenging due to the hierarchical structure inherent in deep neural networks. Instead, this work examines layer-wise negative transfer in the form of feature drift. Let $f_k^l \left( W_k \right)$ be the representation produced by the layer $l$ of the model $W$ on the samples of task $k$. Then, the layer-specific feature drift caused is formulated as:

$$\Delta f_k^l = f_k^l \left( W_{\text{pre}} + T^l \right) - f_k^l \left( W_k \right). \tag{4}$$

where $W_{\text{pre}} + T^l$ denotes the model parameters with only the $l$-th layer merged, i.e., $W_{\text{pre}} + T^l = \{W_{\text{pre}}^1, \dots, W_{\text{pre}}^{l-1}, W_{\text{pre}}^l + T^l, W_{\text{pre}}^{l+1} \dots, W_{\text{pre}}^L\}$.

Sun et al. [29] have derived that the feature drift of every layer contributes to the overall negative transfer, yielding the following upper bound of knowledge conflict:

$$|\Delta \mathcal{L}_k| \leq \beta \sum_{l=1}^{L} \Big( \prod_{m=l+1}^{L} \gamma_m \Big) \|\Delta f_k^l\|. \tag{5}$$

where $\mathcal{L}_k$ is assumed $\beta$-Lipschitz continuous with respect to the network's final output, and the function implemented by each layer $l$ is $\gamma_l$-Lipschitz continuous with respect to its input (i.e., the output of layer $l-1$) within the merging region. For a detailed proof, please refer to Section B.

## 4 Method

Motivated by Eq. (5), our approach aims to mitigate knowledges conflicts by minimizing the feature drift $\|\Delta f_k^l\|$ for each layer $l$. Specifically, we pursue an optimal shared task vector $T^{l^\star}$ by solving the following optimization problem:

$$T^{l^\star} = \arg\min_{T^l} \sum_{k=1}^{K} \|\Delta f_k^l\|^2 = \arg\min_{T^l} \sum_{k=1}^{K} \|f_k^l(W_{\text{pre}} + T^l) - f_k^l(W_k)\|^2. \tag{6}$$

The computation of $T^{l^\star}$ depends on the specific form of $f^l(\cdot)$. In transformer-based architectures, all such operations can be categorized into the following three types:

- **Matrix multiplication**, corresponding to the weight parameters of linear layers;
- **Element-wise (Hadamard) products**, corresponding to the scaling factors in normalization layers;
- **Element-wise addition**, corresponding to the bias parameters of linear layers and the shifting factors in normalization layers.

In our analysis, we treat each minimal computational unit independently. For example, the weight and bias components of a linear layer are considered as separate layers to facilitate analysis. Similarly, in complex modules such as attention blocks, the computations of queries, keys, and values (QKV) are disentangled and analyzed as independent layers. It is also worth noting that the convolutional operation can be equivalently expressed as matrix multiplication [31]. Therefore, we do not treat it as a separate category in our analysis. In the following, we discuss these three types of operations and present the closed-form solution $T^{l^\star}$ for each case.

### 4.1 Solution for Matrix Multiplication

Suppose $W^l \in \mathbb{R}^{d_l \times d_{l+1}}$ corresponds to the weight of a linear layer, which transforms features through matrix multiplication. Specifically, for task $k$, given the input feature matrix $X_k^l \in \mathbb{R}^{n \times d_l}$ extracted by the first $l-1$ layers, the transformed feature representation is expressed as:

$$f_k^l(W) = X_k^l W^l. \tag{7}$$

Substituting Eq. (7) into Eq. (6) yields the following objective:

$$
\begin{aligned}
T^{l\star} &= \arg\min_{T^l} \sum_{k=1}^{K} \|X_k^l(W_{\text{pre}}^l + T^l) - X_k^l(W_k^l)\|_F^2 \\
&= \arg\min_{T^l} \sum_{k=1}^{K} \|X_k^l(T^l - T_k^l)\|_F^2 = \arg\min_{T^l} \sum_{k=1}^{K} \text{trace}((T^l - T_k^l)^\top X_k^{l\top} X_k^l(T^l - T_k^l)).
\end{aligned}
\tag{8}
$$

This defines a convex quadratic optimization problem. Consequently, the optimal solution $T^{l\star}$ can be derived in closed form as follows:

$$
T^{l\star} = \left(\sum_k X_k^{l\top} X_k^l\right)^\dagger \sum_k X_k^{l\top} X_k^l T_k^l,
\tag{9}
$$

where $\dagger$ denotes the Moore–Penrose inverse [9].

## 4.2 Solution for Element-Wise (Hadamard) Multiplication

Next, consider the case where $W^l \in \mathbb{R}^{d_l}$ represents the scaling factors in a normalization layer, i.e.,

$$
f_k^l(W) = X_k^l \circ W^l = [\dots, x_k^l \circ W^l, \dots]^\top,
\tag{10}
$$

where $\circ$ denotes the element-wise product and $x_k^l$ represents a sample feature in $X_k^l$. Under this setting, the objective can be formulated as

$$
T^{l\star} = \arg\min_{T^l} \sum_{k=1}^{K} \sum_{x_k^l} \|x_k^l \circ (W_{\text{pre}}^l + T^l) - x_k^l \circ (W_k^l)\|^2 = \arg\min_{T^l} \sum_{k=1}^{K} \sum_{x_k^l} \sum_{d=1}^{d_l} x_k^l[d]^2 (T^l[d] - T_k^l[d])^2,
\tag{11}
$$

where $x[d]$ represents the $d$-th dimension of $x$. By setting the derivative of the objective with respect to $T^l[d]$ to zero, we obtain the closed-form solution:

$$
T^{l\star}[d] = \frac{\sum_{k=1}^{K} \sum_{x_k^l} x_k^l[d]^2 T_k^l[d]}{\sum_{k=1}^{K} \sum_{x_k^l} x_k^l[d]^2}.
\tag{12}
$$

## 4.3 Solution for Element-Wise Addition

Suppose $W^l \in \mathbb{R}^{d_l}$ represents the bias coefficients, which are added element-wise to $X_k^l$:

$$
f_k^l(W) = X_k^l + W^l = [\dots, x_k^l + W^l, \dots]^\top.
\tag{13}
$$

Thus, the optimal solution can be derived as follows:

$$
T^{l\star} = \arg\min_{T^l} \sum_{k=1}^{K} \sum_{x_k^l} \|x_k^l + (W_{\text{pre}}^l + T^l) - (x_k^l + W_k^l)\|^2 = \arg\min_{T^l} \sum_{k=1}^{K} \|T^l - T_k^l\|^2 = \frac{1}{K} \sum_{k=1}^{K} T_k^l.
\tag{14}
$$

Now, we have derived the optimal vectors for all three types of operation. As summarized in Algorithm 1, we first perform a forward pass over a small exemplar set (16–64 samples per task) to extract each layer's input features for every expert. After obtaining $T^\star$, we integrate it into the pre-trained parameters using a predefined weight $\lambda$:

$$
W_{\text{mtl}}^{\text{lot}} = W_{\text{pre}} + \lambda T^\star.
\tag{15}
$$

Empirically, setting $\lambda = 1$ already yields competitive performance. However, further tuning of $\lambda$ on a validation set, similar to the approach in Task Arithmetic, can lead to additional improvements (see Section 6.3 for a detailed sensitivity analysis).

# 5 Discussion

## 5.1 Exploring the Mechanism of LOT Merging

We present a theoretical analysis of the LOT Merging mechanism, focusing on the behavior of merging matrix-multiplication parameters. Specifically, we apply singular value decomposition (SVD) to the input feature matrices at a given layer $l$, where the representation for task $k$ is given by $X_k^l = U_k^l \Sigma_k^l V_k^{l\top}$. To gain insight into the behavior of LOT Merging, we analyze two extreme cases: (1) the *ideal case* in which task-specific features are mutually orthogonal, and (2) the *worst case* where task-specific features are fully collinear.

**Ideal case**. Assume that for any pair of distinct tasks $k \neq j$, their right singular vectors satisfy the orthogonality condition $V_k^{l\top} V_j^l = 0$. Under this condition, the optimal solution in Eq. (9) simplifies to a summation of the task vectors:

$$T_{\text{ideal}}^{l\star} = \left(\sum_k V_k^l \Sigma_k^{l\,2} V_k^{l\top}\right)^\dagger \sum_k V_k^l \Sigma_k^{l\,2} V_k^{l\top} T_k^l = \sum_k \left(V_k^l \Sigma_k^{l\,2} V_k^{l\top}\right)^\dagger \sum_k V_k^l \Sigma_k^{l\,2} V_k^{l\top} T_k^l = \sum_k V_k^l V_k^{l\top} T_k^l. \tag{16}$$

Here, the term $V_k^l V_k^{l\top} T_k^l$ corresponds to a projection of $T_k^l$ onto the subspace spanned by the singular vectors $V_k^l$. This projection retains only the components of $T_k^l$ aligned with the corresponding feature space $X_k^l$, effectively filtering out irrelevant directions [31]. As a result, LOT Merging introduces no layer-wise feature drift:

$$\sum_{k=1}^{K} \|X_k^l(T_{\text{ideal}}^{l\star} - T_k^l)\|_F^2 = \sum_{k=1}^{K} \|U_k^l \Sigma_k^l V_k^{l\top}(\sum_j V_j^l V_j^{l\top} T_j^l - T_k^l)\|_F^2$$
$$= \sum_{k=1}^{K} \|U_k^l \Sigma_k^l V_k^{l\top} V_k^l V_k^{l\top} T_k^l - U_k^l \Sigma_k^l V_k^{l\top} T_k^l\|_F^2 = 0. \tag{17}$$

**Worst case.** Now we assume the opposite extreme condition that all task features share a common group of singular vectors, i.e., $V_k^l = V^l, \forall k$. Under this condition, the optimal solution in Eq. (9) becomes a weighted average within the shared subspace spanned by $V^l$:

$$T_{\text{worst}}^{l\star} = \left(\sum_k V^l \Sigma_k^{l\,2} V^{l\top}\right)^\dagger \sum_k V^l \Sigma_k^{l\,2} V^{l\top} T_k^l$$

$$= V^l \left(\sum_k \Sigma_k^{l\,2}\right)^\dagger V^{l\top} \sum_k V^l \Sigma_k^{l\,2} V^{l\top} T_k^l = \sum_k \left(V^l \underbrace{\left(\sum_k \Sigma_k^{l\,2}\right)^\dagger \Sigma_k^{l\,2}}_{\text{Normalized Weight}} V^{l\top} T_k^l\right). \tag{18}$$

While feature drift is unavoidable in this setting due to collinearity, the formulation ensures that the deviation is minimized through weighted averaging.

These two extreme cases highlight the adaptability of LOT Merging. In realistic settings where task representations are partially aligned, LOT Merging balances task specificity with shared structure, projecting task-specific transformations into relevant subspaces while aggregating shared patterns. This flexibility allows it to perform robustly across a range of multi-task learning scenarios.

## 5.2 Task Vector Merging vs. Parameter Merging

In this work, we aim to derive an optimal task vector for merging. A natural question arises: why not directly solve for the optimal merging parameters as done in [13]? To illustrate this, consider the direct merging of linear weights via the following objective:

Table 1: Multi-task performance when merging ViT-B/32 models on eight vision tasks. The best performance among training-free methods is highlighted with **bold**. The "#best" column represents the number of datasets where the training-free method performs the best.

| Method | SUN397 | Cars | RESISC45 | EuroSAT | SVHN | GTSRB | MNIST | DTD | Avg Acc | #best |
|---|---|---|---|---|---|---|---|---|---|---|
| *Basic baseline methods* | | | | | | | | | | |
| Pre-trained | 62.3 | 59.7 | 60.7 | 45.5 | 31.4 | 32.6 | 48.5 | 43.8 | 48.0 | - |
| Individual | 75.3 | 77.7 | 96.1 | 99.7 | 97.5 | 98.7 | 99.7 | 79.4 | 90.5 | - |
| Traditional MTL | 73.9 | 74.4 | 93.9 | 98.2 | 95.8 | 98.9 | 99.5 | 77.9 | 88.9 | - |
| *Training-free methods* | | | | | | | | | | |
| Weight Averaging | 65.3 | 63.4 | 71.4 | 71.7 | 64.2 | 52.8 | 87.5 | 50.1 | 65.8 | 0 |
| Fisher Merging | **68.6** | **69.2** | 70.7 | 66.4 | 72.9 | 51.1 | 87.9 | 59.9 | 68.3 | 2 |
| RegMean | 65.3 | 63.5 | 75.6 | 78.6 | 78.1 | 67.4 | 93.7 | 52.0 | 71.8 | 0 |
| Task Arithmetic | 55.2 | 54.9 | 66.7 | 78.9 | 80.2 | 69.7 | 97.3 | 50.4 | 69.1 | 0 |
| Ties-Merging | 59.8 | 58.6 | 70.7 | 79.7 | 86.2 | 72.1 | 98.3 | 54.2 | 72.4 | 0 |
| TATR | 62.7 | 59.3 | 72.3 | 82.3 | 80.5 | 72.6 | 97.0 | 55.4 | 72.8 | 0 |
| Ties-Merging & TATR | 66.3 | 65.9 | 75.9 | 79.4 | 79.9 | 68.1 | 96.2 | 54.8 | 73.3 | 0 |
| Consensus Merging | 65.7 | 63.6 | 76.5 | 77.2 | 81.7 | 70.3 | 97.0 | 57.1 | 73.6 | 0 |
| AWD Merging | 63.5 | 61.9 | 72.6 | 84.9 | 85.1 | 79.1 | 98.1 | 56.7 | 75.2 | 0 |
| PCB Merging | 63.8 | 62.0 | 77.1 | 80.6 | 87.5 | 78.5 | **98.7** | 58.4 | 75.8 | 1 |
| CAT Merging | 68.1 | 65.4 | 80.5 | 89.5 | 85.5 | 78.5 | 98.6 | 60.7 | 78.3 | 0 |
| LOT Merging (ours) | 67.7 | 67.5 | **85.7** | **94.9** | **93.4** | **89.8** | **98.7** | **63.6** | **82.7** | 6 |

$$W^{l\star} = \arg\min_{W^l} \sum_{k=1}^{K} \|X_k^l W^l - X_k^l W_k^l\|_F^2 = \left(\sum_k X_k^{l\top} X_k^l\right)^{\dagger} \sum_k X_k^{l\top} X_k^l W_k^l. \qquad (19)$$

While Eq. (19) provides a straightforward solution, it implicitly modifies the knowledge contained in the pre-trained weights. To make this explicit, we establish a connection with our proposed $T^{l\star}$:

$$W^{l\star} = \underbrace{\left(\sum_k X_k^{l\top} X_k^l\right)^{\dagger} \sum_k X_k^{l\top} X_k^l W_{\text{pre}}^l}_{\text{Modifying Pre-trained Knowledge}} + \underbrace{\left(\sum_k X_k^{l\top} X_k^l\right)^{\dagger} \sum_k X_k^{l\top} X_k^l T_k^l}_{T^{l\star}}. \qquad (20)$$

This decomposition highlights a critical issue: when $X_k^l$ is rank-deficient—which often occurs in practice due to limited exemplar data—the projection term tends to discard useful pre-trained knowledge. The resulting distortion can lead to *catastrophic forgetting* [6] of previously learned representations. To quantify this effect, we compare the accuracy of merged models using $W^{l\star}$ against individual expert models under varying exemplar budgets. As shown in Figure 2, performance degrades significantly as the number of exemplars decreases, where the majority of this degradation stems from the alteration of the pre-trained weights.

Consequently, methods based on Eq. (19), such as [13], typically require large exemplar sets (e.g., 1600 samples per task) to maintain competitive performance [7]. In contrast, our method achieves superior accuracy with as few as 16 to 64 exemplars per task. This substantial reduction in data requirement highlights the effectiveness of our merging strategy in preserving pre-trained knowledge while flexibly adapting to new tasks.

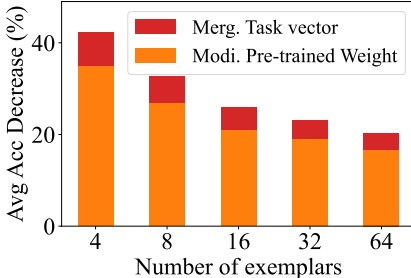

Figure 2: Performance (Avg ACC %) degradation of direct parameter merging compared to Individual under different numbers of exemplars when merging ViT-L/14 models. The error is decomposed into the impact of merging task vectors (red) and modifying pre-trained weights (orange).

Table 2: Multi-task performance when merging ViT-L/14 models on eight vision tasks.

| Method | SUN397 | Cars | RESISC45 | EuroSAT | SVHN | GTSRB | MNIST | DTD | Avg Acc | #best |
|---|---|---|---|---|---|---|---|---|---|---|
| *Basic baseline methods* | | | | | | | | | | |
| Pre-trained | 66.8 | 77.7 | 71.0 | 59.9 | 58.4 | 50.5 | 76.3 | 55.3 | 64.5 | - |
| Individual | 82.3 | 92.4 | 97.4 | 100.0 | 98.1 | 99.2 | 99.7 | 84.1 | 94.2 | - |
| Traditional MTL | 80.8 | 90.6 | 96.3 | 96.3 | 97.6 | 99.1 | 99.6 | 84.4 | 93.5 | - |
| *Training-free methods* | | | | | | | | | | |
| Weight Averaging | 72.1 | 81.6 | 82.6 | 91.9 | 78.2 | 70.7 | 97.1 | 62.8 | 79.6 | 0 |
| Fisher Merging | 69.2 | **88.6** | 87.5 | 93.5 | 80.6 | 74.8 | 93.3 | 70.0 | 82.2 | 1 |
| RegMean | 73.3 | 81.8 | 86.1 | 97.0 | 88.0 | 84.2 | 98.5 | 60.8 | 83.7 | 0 |
| Task Arithmetic | 73.9 | 82.1 | 86.6 | 94.1 | 87.9 | 86.7 | 98.9 | 65.6 | 84.5 | 0 |
| Ties-Merging | 76.5 | 85.0 | 89.3 | 95.7 | 90.3 | 83.3 | 99.0 | 68.8 | 86.0 | 0 |
| TATR | 74.6 | 83.7 | 87.6 | 93.7 | 88.6 | 88.1 | 99.0 | 66.8 | 85.3 | 0 |
| Ties-Merging & TATR | 76.3 | 85.3 | 88.8 | 94.4 | 90.8 | 88.7 | 99.2 | 68.8 | 86.5 | 0 |
| Consensus Merging | 75.0 | 84.3 | 89.4 | 95.6 | 88.3 | 82.4 | 98.9 | 68.0 | 85.2 | 0 |
| AWD Merging | 76.2 | 85.4 | 88.7 | 96.1 | 92.4 | 92.3 | 99.3 | 69.4 | 87.5 | 0 |
| PCB Merging | 76.2 | 86.0 | 89.6 | 95.9 | 89.9 | 92.3 | 99.2 | 71.4 | 87.6 | 0 |
| CAT Merging | **78.7** | 88.5 | 91.1 | 96.3 | 91.3 | **95.7** | 99.4 | 75.7 | 89.6 | 2 |
| LOT Merging (ours) | 76.7 | **88.6** | **91.7** | **98.7** | **97.1** | **95.7** | **99.5** | **76.4** | **90.5** | 7 |

Table 3: Multi-task performance when merging BLIP models on six vision-language tasks.

| Method | COCO Caption | Flickr30k Caption | Textcaps | OKVQA | TextVQA | ScienceQA | #best |
|---|---|---|---|---|---|---|---|
| Metric | CIDEr | CIDEr | CIDEr | Accuracy | Accuracy | Accuracy | |
| Pre-trained · | 0.07 | 0.03 | 0.05 | 42.80 | 21.08 | 40.50 | - |
| Individual | 1.17 | 0.65 | 0.65 | 50.84 | 29.79 | 76.89 | - |
| Task Arithmetic | 0.86 | 0.50 | 0.39 | 17.71 | 0.49 | 40.10 | 0 |
| Ties-Merging | 0.53 | 0.27 | 0.22 | 27.95 | 0.57 | 40.35 | 0 |
| TATR | 0.46 | 0.31 | 0.21 | 28.30 | 14.74 | 42.98 | 0 |
| PCB Merging | 0.71 | 0.52 | 0.30 | 36.04 | 1.88 | 43.01 | 0 |
| CAT Merging | **0.91** | 0.53 | 0.36 | **44.07** | 19.69 | 46.36 | 2 |
| LOT Merging (ours) | **0.91** | **0.54** | **0.44** | 38.35 | **20.82** | **48.24** | 5 |

# 6 Experiments

## 6.1 Settings

**Benchmarks.** Our experiments cover vision and vision-language tasks. For the vision tasks, we follow [12] and utilize eight image classification datasets: SUN397 [36], Cars [14], RESISC45 [2], EuroSAT [8], SVHN [22], GTSRB [28], MNIST [15], and DTD [3]. For the vision-language tasks, we focus on three captioning datasets (COCO Caption [1], Flickr30k Caption [23], Textcaps [26]) and three Visual Question Answering (VQA) datasets (OKVQA [19], TextVQA [27], and ScienceQA [17]).

**Baselines.** We select several training-free model merging methods as the primary comparison baselines, including weight averaging, Fisher Merging [20], RegMean [13], Task Arithmetic [12], Ties-Merging [38], TATR [30], Ties-Merging & TATR [30], Consensus Merging [34], AWD Merging [37], PCB Merging [5], and CAT Merging [29]. We also present three baseline methods for reference: Pre-trained model performance, Individual fine-tuned model performance, and Traditional Multi-Task Learning (MTL) performance.

**Metrics.** For both classification and VQA tasks, we employ accuracy as the evaluation metric. For captioning tasks, we select CIDEr as the evaluation criterion. To minimize potential performance fluctuations arising from the selection of different exemplar sets, we rerun LOT Merging with three randomly selected exemplar sets and report the average performance.

**Implementation details.** For vision tasks, following [12], we utilize the vision encoder of CLIP [24] as the pre-trained model, including both ViT-B/32 and ViT-L/14 versions. The task vectors are obtained from the official repository of [12]. When merging vision-language tasks, task vectors are generated by fine-tuning the VQA version of BLIP [25] for 6000 steps per task. The BLIP architecture consists of an image encoder, a text encoder, and a text decoder, with all model weights fine-tuned during training. Further details of the experiments can be found in our supplementary appendix and code. All exemplar sets are randomly sampled from the training set and are kept strictly separate from the test set used for evaluation.

## 6.2 Comparison Results

**Merging ViT-B/32 models on vision tasks.** We first compare multi-task performance when merging ViT-B/32 models on eight vision tasks (Table 1). Among the training-free methods, Fisher Merging achieves the highest accuracy on SUN397 and Cars, while our approach performs the best on all the remaining six tasks. Notably, our approach achieves the highest average accuracy of 82.7%, suppressing the second-best training-free methods with 4.4%.

**Merging ViT-L/14 models on vision tasks.** Next, we evaluate the multi-task performance when merging ViT-L/14 models on eight vision tasks (Table 2). Our method also achieves the highest average accuracy of 90.5%, higher than the second-best performance by 0.9%. Furthermore, our method performs best in seven tasks, confirming its robustness.

**Merging vision-language tasks.** Finally, we compare multi-task performance across six vision-language tasks when merging BLIP models (Table 3). Our method leads the performance with the highest CIDEr scores on COCO Caption, Flickr30k Caption, and Textcaps, as well as the highest accuracy on TextVQA and ScienceQA. In total, our method delivers the best performance in five tasks, showcasing its adaptability in vision-language tasks.

## 6.3 Sensitivity Analysis

**Sensitivity analysis of exemplar number.** The number of exemplars impacts LOT Merging by influencing the accuracy of $X^{l\top}X^l$. As shown in Figure 3 (a), the performance of LOT Merging steadily improves as the number of exemplars increases. Notably, when the number of samples reaches only 16 per task, performance stabilizes with respect to the exemplar number. This demonstrates the robustness of LOT Merging in data-less scenarios (e.g., 16 samples per task). Based on our experiments, we empirically recommend using 64 samples per task.

**Sensitivity analysis of $\lambda$ (scaling factor in Eq. (15)).** The scaling factor $\lambda$ controls the contribution of task vectors to the merged model. As illustrated in Figure 3 (b), performance remains relatively stable for $\lambda$ values in the range of 1.0 to 1.5. For larger

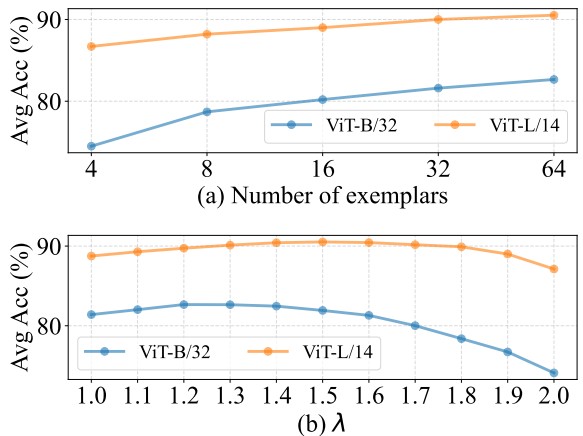

Figure 3: Average accuracy (%) on eight vision tasks with various numbers of exemplars (a) and $\lambda$ (scaling factor in Eq. (15)).

values of $\lambda$, performance gradually decreases. Based on our empirical analysis, we set $\lambda$ to 1.2 for merging ViT-B/32 and 1.5 for merging ViT-L/14.

## 7 Conlusion

In this paper, we propose Layer-wise Optimal Task Vector Merging (LOT Merging), a novel approach to mitigate the problem of feature drift in model merging. By adaptively merging task-specific knowledge at the layer level, LOT Merging reorganizes the merging process as an optimization task that minimizes the discrepancy between the representations of individual models before and after merging. This adaptive strategy ensures that task-specific information is retained while minimizing interference across tasks. LOT Merging requires no retraining and is suitable for scenarios with limited samples, making it highly efficient and applicable in resource-constrained environments. Our experiments on both vision and vision-language benchmarks demonstrate that LOT Merging significantly outperforms baseline methods, achieving up to 4.4% improvement on vision tasks and showing strong performance across various vision-language tasks.

While LOT Merging achieves strong empirical results with minimal data, it still requires access to a small exemplar set, which may not be feasible in strictly data-free scenarios. In addition,

although the method supports mainstream transformer architectures, extending it to models with more complex operations (e.g., layers involving exponential functions) may require additional adaptation. Future work could explore how feature drift interacts across layers to better disentangle conflicting knowledge and broaden the applicability of the approach.

## Acknowledgments

This work was supported, in part, by the Fundamental Research Funds for the Central Universities under Grants 2025JBRC004, 2023JBZY037, and 2022JBQY007, by the Beijing Natural Science Foundation under Grant L231019, by the Hebei Natural Science Foundation under Grant F2025106045, by the National Natural Science Foundation of China under Grants 62276019, 62306028, 62501043, and U22B2004, by the Shenzhen Science and Technology Program Project under Grant KJZD20240903102742055, by the RIE2025 Industry Alignment Fund – Industry Collaboration Projects (IAF-ICP) (Award I2301E0026), administered by A*STAR, by Alibaba Group and NTU Singapore through Alibaba-NTU Global e-Sustainability CorpLab (ANGEL), and by the Nanyang Associate Professorship and National Research Foundation Fellowship (NRFF13-2021-0006), Singapore.

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

# A Pseudo Code

Algorithm 1 outlines the proposed LOT merging process. Given a pre-trained model and a set of task vectors, we first extract input features from exemplar sets using task-specific adapters. Then, for each layer, we compute the optimal merged task vector by minimizing the discrepancy between the adapted and reference features. The final model is obtained by linearly combining the pre-trained weights with the optimized task vectors.

---

**Algorithm 1:** The model merging process

---

**Input:** Pre-trained model $W_{\text{pre}}$; Task vectors $\{T_1, \ldots, T_K\}$; Exemplar set $\{M_1, \ldots, M_K\}$
**Output:** Merged model $W_{\text{mtl}}^{\text{lot}}$

1  $T^\star = \{T^{1^\star}, \ldots, T^{L^\star}\}$ // Initial
   // Collect the input features
2  **for** $k = 1$ *to* $K$ **do**
3     Initialize task inputs: $X_k^1 = M_k$
4     **for** $l = 1$ *to* $L$ **do**
5        $X_k^{l+1} = f(X_k^l; W_0^l + T_k^l)$

   // Compute optimal task vector for each layer
6  **for** $l = 1$ *to* $L$ **do**
   // Refer to (9), (12) & (14).
7     $T^{l^\star} = \arg\min_{T^l} \sum_{k=1}^{K} \|f_k^l(W_{\text{pre}}^l + T^l) - f_k^l(W_{\text{pre}}^l + T_k^l)\|^2$
8  **// Merging**
9  $W_{\text{mtl}}^{\text{lot}} = W_{\text{pre}} + \lambda T^\star$
10  **return** $W_{\text{mtl}}^{\text{lot}}$

---

# B Proof of Eq. (5)

Let $W_k$ be the parameters for task $k$, and let $T$ be a task vector. The feature dirt at layer $l$ caused by $T_i$ is defined as:

$$\Delta f_k^l = f_k^l(W_{pre} + T^l) - f_k^l(W_k).$$

We aim to prove the following bound:

$$|\Delta \mathcal{L}_k| \leq \beta \sum_{l=1}^{L} \Big( \prod_{m=l+1}^{L} \gamma_m \Big) \|\Delta f_k^l\|, \tag{21}$$

where $\mathcal{L}$ is assumed $\beta$-Lipschitz continuous with respect to the network's final output, and each layer $l$ is $\gamma_l$-Lipschitz continuous with respect to its input (i.e., the output of layer $l-1$) within the merging region.

*Proof.* The feature shift at layer $l$ can be decomposed as:

$$
\begin{aligned}
f_k^l(W_{pre} + T) - f_k^l(W_k) &= f_k^l(f_k^{l-1}(W_{pre} + T); W_{pre} + T^l) - f_k^l(f_k^{l-1}(W_k); W_k) \\
&= f_k^l(f_k^{l-1}(W_{pre} + T); W_{pre} + T^l) - f_k^l(f_k^{l-1}(W_k); W_{pre} + T^l) \\
&+ f_k^l(f_k^{l-1}(W_k); W_{pre} + T^l) - f_k^l(f_k^{l-1}(W_k); W_k).
\end{aligned}
\tag{22}
$$

Under the norm-induced triangular inequality, we have:

$$
\begin{aligned}
\|f_k^l(W_{pre} + T) - f_k^l(W_k)\| &\leq \|f_k^l(f_k^{l-1}(W_{pre} + T); W_{pre} + T^l) - f_k^l(f_k^{l-1}(W_k); W_{pre} + T^l)\| \\
&+ \|f_k^l(f_k^{l-1}(W_k); W_{pre} + T^l) - f_k^l(f_k^{l-1}(W_k); W_k)\|.
\end{aligned}
\tag{23}
$$

- For the first part, we apply the $\gamma_l$-Lipschitz continuity of $f_k^l$:

$$\|f_k^l(f_k^{l-1}(W_{pre}+T); W_{pre}+T^l) - f_k^l(f_k^{l-1}(W_k); W_{pre}+T^l)\| \leq \gamma_l \|f_k^{l-1}(W_{pre}+T) - f_k^{l-1}(W_k)\|.$$

- For the second part:

$$\|f_k^l(f_k^{l-1}(W_k); W_{pre} + T^l) - f_k^l(f_k^{l-1}(W_k); W_k)\| = \|\Delta f_k^l\|.$$

Combining these two terms, we have:

$$\|f_k^l(W_{pre} + T) - f_k^l(W_k)\| \leq \gamma_l \|f_k^{l-1}(W_{pre} + T) - f_k^{l-1}(W_k)\| + \|\Delta f_k^l\|.$$

Unfolding this recursive inequality from $l = 1$ to $l = L$ and accumulating the error gives:

$$\|f_k^L(W_{pre} + T) - f_k^L(W_k)\| \leq \sum_{l=1}^{L} \Big( \prod_{m=l+1}^{L} \gamma_m \Big) \|\Delta f_k^l\|. \tag{24}$$

Using the assumption that $\mathcal{L}$ is $\beta$-Lipschitz continuous with respect to the network's final output, we conclude:

$$|\Delta \mathcal{L}_k| \leq \beta \sum_{l=1}^{L} \Big( \prod_{m=l+1}^{L} \gamma_m \Big) \|\Delta f_k^l\|. \tag{25}$$

$\square$

# C   Experimental Setup

This section provides an overview of the experimental setup, including details about the computational environment, datasets, and the baseline models employed in the experiments.

## C.1   Computational Resources

All experiments described in this paper were performed on a workstation running Ubuntu 16.04. The system configuration includes dual Intel Xeon 2.60GHz CPUs, 256 GB of RAM, and six NVIDIA RTX 3090 GPUs. The code was implemented in Python 3.8 and executed on this hardware platform to ensure consistency across all experiments.

## C.2   Datasets

Our experimental procedure follows the guidelines outlined in Task Arithmetic [12], utilizing eight commonly used image classification datasets, which are summarized below:

- **SUN397** [36]: A large-scale dataset comprising 108,754 images, organized into 397 categories. Each category contains a minimum of 100 images, making this dataset a comprehensive benchmark for scene classification tasks.
- **Stanford Cars** [14]: A fine-grained dataset with 16,185 images of 196 distinct car models. The dataset is split evenly into training and testing sets, providing a reliable resource for evaluating car model recognition systems.
- **RESISC45** [2]: This remote sensing dataset consists of 31,500 images representing 45 different scene categories. Each category contains around 700 images, covering a broad range of geographical and structural themes.
- **EuroSAT** [8]: A satellite image dataset containing 27,000 labeled and geo-referenced images. The dataset is divided into 10 categories, including forests, urban areas, and agricultural fields, designed for land-use classification tasks.
- **SVHN** [22]: This dataset, derived from real-world street view images, consists of 73,257 training and 26,032 test images of digits, distributed across 10 classes. An additional 531,131 samples are included for extended training purposes.

- **GTSRB** [28]: The German Traffic Sign Recognition Benchmark, which includes over 50,000 images across 43 traffic sign categories. This dataset is a well-known benchmark for traffic sign recognition systems.
- **MNIST** [15]: A foundational dataset for handwritten digit classification, consisting of 60,000 training images and 10,000 test images, distributed across 10 digit classes.
- **DTD** [3]: A dataset designed for texture classification, containing 5,640 images across 47 categories, with approximately 120 images per category. It is commonly used for evaluating texture recognition algorithms.

In addition, we also utilize six vision-language datasets, which are detailed below:

- **COCO Caption** [1]: A large image captioning dataset derived from the MS COCO collection. It contains over 330,000 images, each annotated with five different captions, aimed at training models to generate natural language descriptions for images.
- **Flickr30k Caption** [23]: This dataset consists of 31,000 images from Flickr, with each image paired with five descriptive sentences. It is used for both image captioning and retrieval tasks.
- **TextCaps** [26]: A challenging dataset for image captioning, where the captions require reasoning over both visual and textual information present within the image. The dataset includes 145,000 image-caption pairs.
- **OKVQA** [19]: A knowledge-based visual question answering dataset that includes over 14,000 questions requiring external knowledge to answer. The dataset is designed to assess reasoning beyond visual content alone.
- **TextVQA** [27]: A visual question answering dataset that emphasizes reading and interpreting text embedded within images. It contains over 45,000 questions across 28,000 images, necessitating both visual and textual reasoning.
- **ScienceQA** [17]: A multi-modal dataset for scientific question answering, consisting of over 21,000 multiple-choice questions paired with images and textual explanations across various scientific disciplines, including biology, chemistry, and physics.

## C.3 Baseline Methods

In this study, we compare our approach with several baseline methods. Below, we provide a description of each baseline:

- **Pre-trained**: This baseline uses a pre-trained model to perform tasks across multiple domains. Since it does not leverage any task-specific fine-tuning, it typically results in suboptimal performance on downstream tasks.
- **Individual**: In this method, separate models are fine-tuned for each task individually. Although this avoids task interference, it is limited by the inability to perform multi-task learning simultaneously. It represents a reference for the best possible performance, or *upper bound*, for merging methods.
- **Traditional MTL**: This approach combines the training data from all tasks and trains a single multi-task model. It serves as a traditional method for joint task learning.

The following are compression-based approaches:

- **EMR Merging** [11]: This method applies lightweight task-specific masks and rescalers to compress task vectors.
- **WEMOE** [32]: Upscales MLP layers into input-dependent Mixture-of-Experts modules to dynamically integrate shared and task-specific knowledge during model merging.

The following are test-time training-based approaches:

- **AdaMerging** [41]: This method adapts the model to new tasks by using an unlabeled test set to learn adaptive merging coefficients at both the task- and layer-level, as applied in Task Arithmetic.

- **AdaMerging++** [41]: An improved version of AdaMerging, which integrates the principles of Ties-Merging [38] to further enhance the adaptive merging process.
- **Surgery** [40]: Surgery introduces a feature transformation module that aligns the features from different tasks during the merging process. In our experiments, we utilize the basic version of Surgery alongside Task Arithmetic for evaluation.
- **Localize-and-Stitch** [7]: This method optimally combines the strengths of several fine-tuned models by identifying and localizing essential regions within each model before merging.

The following methods do not require training:

- **Weight Averaging**: This technique averages the parameters from the models of different tasks to form a single multi-task model without any additional training steps.
- **Fisher Merging** [20]: This approach uses the Fisher information matrix to assess the relative importance of model parameters, merging them based on their significance.
- **RegMean** [13]: RegMean adjusts and combines rows from weight matrices based on statistical information gathered from the training data, refining the model's parameters.
- **Task Arithmetic** [12]: Task Arithmetic introduces the concept of a "task vector," which is defined as the difference between fine-tuned model parameters and pre-trained model parameters. By combining multiple task vectors and adding them to the pre-trained model, it enables multi-task learning.
- **Ties-Merging** [38]: Ties-Merging removes insignificant parameters from the task vectors and resolves sign conflicts, reducing interference during the merging of task vectors.
- **TATR** [30]: TATR improves upon Task Arithmetic by restricting the merging of task vectors to a defined trust region, which reduces knowledge conflicts between tasks.
- **TATR & Ties-Merging** [30, 38]: This method combines the trust region restriction of TATR with Ties-Merging to further enhance task vector merging.
- **Consensus Merging** [34]: This method computes a set of masks for each task vector to minimize the distance between the merged model and each fine-tuned model in the parameter space.
- **AWD Merging** [37]: AWD Merging generates redundant vectors such that subtracting them from the original task vectors leads to increased orthogonality in the remaining vectors.
- **PCB Merging** [5]: PCB Merging trims components of the task vector that have small magnitudes and are not strongly correlated with other tasks.
- **CAT Merging** [29]: CAT Merging trims components (using projection or mask) of the task vector that may cause knowledge conflicts.

# D  Additional Experiments

## D.1  Additional Performance Comparision

In this section, we compare LOT Merging with both compression-based methods and test-time adaptation-based approaches. Compression-based methods rely on techniques such as masking [11] or singular value decomposition (SVD) [32] to compress task vectors, and typically require manually specified or trained routers during inference to select the appropriate task vector. On the other hand, test-time adaptation-based methods leverage unlabeled test data to train merging weights or certain modules, which introduces additional computational overhead and data requirements.

It is important to note that comparing LOT Merging with these methods is inherently **unfair**, as LOT Merging is a training-free approach and does not require any adaptation or auxiliary modules at inference time. Moreover, the merged model of LOT Merging retains the original network architecture without introducing any structural modifications. As shown in Table 4, while LOT Merging lags behind compression-based methods in terms of average accuracy, it achieves comparable or even superior performance to several representative test-time adaptation approaches. This highlights the strength of LOT Merging in competitive performance across diverse vision tasks.

Table 4: Multi-task performance when merging on eight vision tasks. Results of Localize-and-Stitch with "†" stem from the original paper, where only the performance on ViT-B/32 is provided.

| Method | SUN397 | Cars | RESISC45 | EuroSAT | SVHN | GTSRB | MNIST | DTD | Avg Acc |
|---|---|---|---|---|---|---|---|---|---|
| *ViT-B/32* | | | | | | | | | |
| *Compression based methods* | | | | | | | | | |
| WEMOE | 74.1 | 77.4 | 93.7 | 99.1 | 96.2 | 98.9 | 99.6 | 76.4 | 89.4 |
| EMR-Merging | 75.2 | 72.8 | 93.5 | 99.5 | 96.9 | 98.1 | 99.6 | 74.4 | 88.7 |
| *Test-time Adaption based methods* | | | | | | | | | |
| TW AdaMerging | 58.0 | 53.2 | 68.8 | 85.7 | 81.1 | 84.4 | 92.4 | 44.8 | 71.1 |
| TW AdaMerging++ | 60.8 | 56.9 | 73.1 | 83.4 | 87.3 | 82.4 | 95.7 | 50.1 | 73.7 |
| LW AdaMerging | 64.5 | 68.1 | 79.2 | 93.8 | 87.0 | 91.9 | 97.5 | 59.1 | 80.1 |
| LW AdaMerging++ | 66.6 | 68.3 | 82.2 | 94.2 | 89.6 | 89.0 | 98.3 | 60.6 | 81.1 |
| Surgery Merging | 63.8 | 59.9 | 83.3 | 97.9 | 87.0 | 87.0 | 98.6 | 69.4 | 80.9 |
| Localize-and-Stitch† | 67.2 | 68.3 | 81.8 | 89.4 | 87.9 | 86.6 | 94.8 | 62.9 | 79.9 |
| *Training-free methods* | | | | | | | | | |
| LOT Merging (ours) | 67.7 | 67.5 | 85.7 | 94.9 | 93.4 | 89.8 | 98.7 | 63.6 | 82.7 |
| *ViT-L/14* | | | | | | | | | |
| *Compression based methods* | | | | | | | | | |
| WEMOE | 81.4 | 92.6 | 95.4 | 99.4 | 97.7 | 99.3 | 99.7 | 83.7 | 93.6 |
| EMR-Merging | 83.2 | 90.7 | 96.8 | 99.7 | 97.9 | 99.1 | 99.7 | 82.7 | 93.7 |
| *Test-time Adaption based methods* | | | | | | | | | |
| AdaMerging | 79.0 | 90.3 | 90.8 | 96.2 | 93.4 | 98.0 | 99.0 | 79.9 | 90.8 |
| AdaMerging++ | 79.4 | 90.3 | 91.6 | 97.4 | 93.4 | 97.5 | 99.0 | 79.2 | 91.0 |
| Surgery Merging | 75.7 | 84.4 | 93.1 | 98.8 | 91.3 | 93.4 | 99.1 | 76.1 | 89.0 |
| *Training-free methods* | | | | | | | | | |
| LOT Merging (ours) | 76.7 | 88.6 | 91.7 | 98.7 | 97.1 | 95.7 | 99.5 | 76.4 | 90.5 |

Table 5: Multi-task performance when merging ViT-B/16 models on eight tasks.

| Method | SUN397 | Cars | RESISC45 | EuroSAT | SVHN | GTSRB | MNIST | DTD | Avg Acc |
|---|---|---|---|---|---|---|---|---|---|
| Pre-trained | 63.8 | 64.6 | 65.7 | 54.5 | 52.0 | 43.3 | 51.7 | 45.1 | 55.0 |
| Individual | 81.8 | 86.8 | 96.9 | 99.7 | 97.8 | 99.1 | 99.7 | 82.0 | 92.9 |
| Weight Averaging | 67.7 | 70.0 | 75.3 | 79.5 | 74.9 | 60.1 | 94.4 | 43.8 | 70.7 |
| Fisher Merging | 68.5 | 69.9 | 75.2 | 80.4 | 73.2 | 61.2 | 94.5 | 50.7 | 71.7 |
| RegMean | 69.1 | 71.6 | 77.6 | 88.8 | 83.7 | 70.2 | 96.9 | 54.6 | 76.6 |
| Task Arithmetic | 61.1 | 65.9 | 74.0 | 76.2 | 88.0 | 73.9 | 98.4 | 53.0 | 73.8 |
| Ties-Merging | 69.1 | 72.5 | 80.5 | 84.0 | 85.0 | 71.5 | 98.1 | 54.9 | 77.0 |
| TATR | 67.4 | 70.4 | 77.9 | 81.7 | 87.6 | 77.2 | 98.3 | 55.6 | 77.0 |
| AWD Merging | 67.8 | 72.7 | 78.7 | 88.5 | 90.9 | 83.6 | 98.9 | 57.1 | 79.8 |
| CAT Merging | **72.9** | 75.9 | 83.1 | 92.8 | 88.2 | 82.7 | 98.8 | 62.7 | 82.1 |
| LOT Merging (ours) | 71.0 | **76.2** | **87.6** | **95.8** | **96.5** | **91.9** | **99.2** | **67.0** | **85.7** |

## D.2 Comparison on ViT-B/16

Table 5 presents the multi-task performance of various model merging techniques on eight diverse tasks using ViT-B/16. As can be seen, LOT Merging outperforms all baselines, achieving the highest average accuracy (85.7%). Notably, it excels in almost all tasks, demonstrating its ability to balance task-specific feature retention and shared information utilization. These results highlight the effectiveness of LOT Merging in multi-task learning scenarios.

## D.3 Analysis of Exemplar Number

Table 6 presents the multi-task performance of LOT Merging with varying numbers of exemplars. The results are shown for both the ViT-B/32 and ViT-L/14 models with different numbers of exemplars used for merging, ranging from 4 to 64. As the number of exemplars increases, we observe a consistent improvement in performance across all datasets for both ViT architectures. Notably, LOT Merging already achieves state-of-the-art performance with only 8 exemplars per task, demonstrating that LOT Merging is particularly well-suited for dataless adaptation settings.

Table 6: Multi-task performance of LOT Merging with various numbers of exemplars. The "#exemplar" column represents the number of exemplars used for merging.

| #exemplar | SUN397 | Cars | RESISC45 | EuroSAT | SVHN | GTSRB | MNIST | DTD | Avg Acc |
|---|---|---|---|---|---|---|---|---|---|
| | | | | *ViT-B/32* | | | | | |
| 4 | 64.2 | 64.3 | 69.1 | 84.4 | 88.6 | 77.9 | 96.5 | 51.1 | 74.5 |
| 8 | 65.4 | 66.3 | 79.4 | 87.6 | 92.8 | 85.5 | 98.2 | 54.1 | 78.7 |
| 16 | 66.4 | 67.1 | 82.6 | 90.9 | 91.7 | 86.1 | 98.5 | 58.1 | 80.2 |
| 32 | 67.4 | 67.5 | 83.3 | 92.4 | 93.2 | 89.6 | 98.6 | 60.6 | 81.6 |
| 64 | 67.7 | 67.5 | 85.7 | 94.9 | 93.4 | 89.8 | 98.7 | 63.6 | 82.7 |
| | | | | *ViT-L/14* | | | | | |
| 4 | 74.1 | 87.5 | 86.6 | 92.6 | 95.3 | 92.1 | 99.1 | 66.5 | 86.7 |
| 8 | 75.4 | 87.8 | 88.4 | 95.9 | 96.4 | 93.9 | 99.3 | 68.3 | 88.2 |
| 16 | 75.7 | 87.6 | 89.0 | 98.2 | 96.6 | 94.0 | 99.4 | 71.3 | 89.0 |
| 32 | 76.4 | 88.4 | 91.2 | 98.3 | 97.0 | 95.5 | 99.5 | 74.1 | 90.0 |
| 64 | 76.7 | 88.6 | 91.7 | 98.7 | 97.1 | 95.7 | 99.5 | 76.4 | 90.5 |

Table 7: Ablation Study of LOT Merging.

| Linear Weight | Scaler | Bias | SUN397 | Cars | RESISC45 | EuroSAT | SVHN | GTSRB | MNIST | DTD | Avg Acc |
|---|---|---|---|---|---|---|---|---|---|---|---|
| | | | | | | *ViT-B/32* | | | | | |
| | ✓ | ✓ | 61.9 | 57.4 | 60.4 | 55.9 | 36.2 | 30.0 | 54.3 | 42.3 | 49.8 |
| ✓ | | ✓ | 67.8 | 67.9 | 85.9 | 92.6 | 93.3 | 87.5 | 98.7 | 62.0 | 81.9 |
| ✓ | ✓ | | 67.9 | 67.7 | 84.6 | 93.5 | 93.2 | 90.5 | 98.2 | 64.0 | 82.4 |
| ✓ | ✓ | ✓ | 67.7 | 65.5 | 85.7 | 94.9 | 93.4 | 89.8 | 98.7 | 63.6 | 82.7 |
| | | | | | | *ViT-L/14* | | | | | |
| | ✓ | ✓ | 67.3 | 78.5 | 72.3 | 64.6 | 59.5 | 50.4 | 76.7 | 55.6 | 65.6 |
| ✓ | | ✓ | 76.7 | 88.4 | 92.4 | 98.4 | 97.0 | 94.8 | 99.5 | 75.3 | 90.3 |
| ✓ | ✓ | | 76.3 | 88.3 | 91.7 | 98.6 | 97.1 | 95.2 | 99.5 | 75.3 | 90.2 |
| ✓ | ✓ | ✓ | 76.7 | 88.6 | 91.7 | 98.7 | 97.1 | 95.7 | 99.5 | 76.4 | 90.5 |

## D.4 Ablation Study

In this ablation study, we investigate the impact of merging pre-trained and learned parameters on the performance of Vision Transformers (ViT). Specifically, we examine three types of parameters: linear weight, scaling factors, and bias coefficient. For each type, we replace the merged parameters with the pre-trained ones and evaluate the effect on model performance.

The results, shown in Table 7, demonstrate that merging each type of parameter leads to consistent improvements across the board. Among the three parameter types, merging the Linear Weight yields the largest performance boost, as it represents the majority of the model's parameters.

Table 8: Multi-task performance of LOT Merging with three random exemplar sets.

| | SUN397 | Cars | RESISC45 | EuroSAT | SVHN | GTSRB | MNIST | DTD | Avg Acc |
|---|---|---|---|---|---|---|---|---|---|
| | | | | *ViT-B/32* | | | | | |
| Random exemplar set 1 | 67.8 | 67.5 | 85.3 | 94.6 | 93.4 | 89.8 | 98.7 | 63.9 | 82.6 |
| Random exemplar set 2 | 67.6 | 67.5 | 85.8 | 94.6 | 93.7 | 89.2 | 98.6 | 63.9 | 82.6 |
| Random exemplar set 3 | 67.7 | 67.7 | 86.1 | 95.6 | 93.3 | 90.5 | 98.7 | 63.1 | 82.8 |
| Average | 67.7 | 67.5 | 85.7 | 94.9 | 93.4 | 89.8 | 98.7 | 63.6 | 82.7 |
| | | | | *ViT-L/14* | | | | | |
| Random exemplar set 1 | 76.7 | 88.8 | 91.5 | 98.7 | 97.0 | 95.7 | 99.6 | 76.3 | 90.5 |
| Random exemplar set 2 | 76.8 | 88.3 | 91.6 | 98.8 | 97.1 | 95.5 | 99.5 | 76.5 | 90.5 |
| Random exemplar set 3 | 76.6 | 88.6 | 91.9 | 98.4 | 97.0 | 95.9 | 99.4 | 76.3 | 90.5 |
| Average | 76.7 | 88.6 | 91.7 | 98.7 | 97.1 | 95.7 | 99.5 | 76.4 | 90.5 |

## D.5 Robustness Analysis of Exemplar Sets

Table 8 demonstrates the robustness of LOT Merging across three random exemplar sets. For both ViT-B/32 and ViT-L/14, the average accuracy remains highly stable ($\leq 0.2\%$ variance), with minimal fluctuations across datasets. The stronger ViT-L/14 backbone further enhances stability, maintaining a consistent $90.5\%$ average accuracy. These results confirm that LOT Merging is highly robust to exemplar selection, ensuring reliable performance across diverse tasks.

Table 9: Computational complexity comparison (in seconds) for merging ViT-B/32 and ViT-L/14 models across eight vision tasks, measured on a single RTX 3090 GPU.

| Method | TA w/ Surgery | AdaMerging | TATR | PCB Merging | CAT Merging | LOT Merging (ours) |
|---|---|---|---|---|---|---|
| ViT-B/32 | 12621 | 8276 | 176 | 43 | 46 | 44 |
| ViT-L/14 | 36826 | 16299 | 283 | 131 | 150 | 161 |

## D.6 Analysis of Computational Complexity

The computational overhead of LOT Merging is reasonable and practically efficient. The procedure consists of two primary stages:

- **Feature extraction:** This step is highly efficient, requiring only a small number of unlabeled examples (typically 16–64 per task). As it avoids training or gradient-based updates, the computational cost remains minimal;

- **Optimal task-vector computation:** This step relies on a closed-form solution, eliminating the need for iterative optimization. For linear task representations, this involves a matrix inversion operation. While matrix inversion can be associated with higher theoretical complexity ($\mathcal{O}(d^3)$), we mitigate this overhead through parallelized implementation on modern GPUs. In practice, the computation is highly efficient and introduces negligible latency.

Empirical runtime measurements, summarized in Table 9 (recorded on a single NVIDIA RTX 3090 GPU), confirm that LOT Merging achieves significantly lower wall-clock time compared to training-based baselines such as TA with Surgery [40] and AdaMerging [41]. These results highlight the practical efficiency of our approach, making it highly suitable for scenarios where rapid adaptation across tasks is required.

Table 10: Consistency comparison for merging ViT-B/32 and ViT-L/14 models across eight vision tasks.

| | Fisher Merging | RegMean | Task Arithmetic | PCB Merging | CAT Merging | LOT Merging(ours) |
|---|---|---|---|---|---|---|
| ViT-B/32 | 13.78 | 8.46 | 8.95 | 6.85 | 6.21 | 4.58 |
| ViT-L/14 | 6.79 | 6.86 | 5.11 | 3.49 | 2.51 | 2.64 |

## D.7 Analysis of Consistency

While LOT Merging achieves the highest average performance in Tables 1 and 2, it does not consistently deliver the best accuracy on every individual dataset. This variation can be attributed to methods like Fisher merging, which tend to yield uneven performance across tasks. To quantify such inconsistency, we compute the standard deviation of accuracy drops—defined as the difference between the task-specific and merged model accuracies—across all tasks. As shown in Table 10, LOT Merging exhibits low variance, indicating a more balanced integration of task knowledge without favoring specific tasks.

Table 11: Results (%) on seen and unseen Tasks

| | SUN397 | Cars | RESISC45 | DTD | SVHN | GTSRB | Avg Acc (Seen) | MNIST (Unseen) | EuroSAT (Unseen) | Avg Acc (Unseen) |
|---|---|---|---|---|---|---|---|---|---|---|
| Task Arithmetic | 63.3 | 62.4 | 75.1 | 57.8 | 84.6 | 80.4 | 70.6 | 77.2 | 46.2 | 61.7 |
| Ties-Merging | 67.8 | 66.2 | 77.2 | 56.7 | 77.1 | 70.9 | 69.3 | 75.9 | 43.3 | 59.6 |
| LOT Merging | 69.1 | 69.1 | 89.7 | 65.6 | 94.1 | 93.7 | 80.2 | 81.5 | 45.4 | 63.5 |

| | SUN397 | Cars | GTSRB | EuroSAT | DTD | MNIST | Avg Acc (Seen) | RESISC45 (Unseen) | SVHN (Unseen) | Avg Acc (Unseen) |
|---|---|---|---|---|---|---|---|---|---|---|
| Task Arithmetic | 64.0 | 64.0 | 75.2 | 87.7 | 57.0 | 95.7 | 73.9 | 52.3 | 44.9 | 51.1 |
| Ties-Merging | 68.0 | 67.1 | 67.7 | 78.4 | 56.5 | 92.8 | 71.8 | 58.7 | 49.2 | 53.9 |
| LOT Merging | 68.8 | 69.9 | 84.8 | 95.8 | 61.5 | 97.5 | 79.7 | 56.3 | 54.6 | 55.5 |

## D.8 Analysis of Generalization Ability

We first evaluated our method by merging models trained on six out of eight tasks and then testing on the remaining two unseen tasks. As shown in the tables 11, LOT Merging consistently outperforms both Task Arithmetic and Ties-Merging—not only on seen tasks, but also on unseen (out-of-domain) tasks. These results demonstrate the moderate generalization ability of our approach. However, the performance gap does become narrower on out-of-domain tasks compared to in-domain tasks.

Table 12: Average accuracy with Gaussian blur kernel

| Method | Gaussian Blur Kernel | ViT-B/32 (%) | ViT-L/14 (%) |
|---|---|---|---|
| Task Arithmetic | – | 69.1 | 84.5 |
| LOT Merging | $kernel\_size = 3, \sigma = 1$ | 82.2 | 90.3 |
| LOT Merging | $kernel\_size = 5, \sigma = 2$ | 80.0 | 89.5 |
| LOT Merging | $kernel\_size = 7, \sigma = 3$ | 78.3 | 88.7 |

Table 13: Performance when using similar distributed exemplars

| Method | COCO Caption | OKVQA | ScienceQA |
|---|---|---|---|
| Task Arithmetic | 0.53 | 40.37 | 51.55 |
| LOT Merging | 0.87 | 44.25 | 60.08 |
| LOT Merging w/ surrogate exemplars | 0.62 | 40.15 | 53.54 |

This observation motivates us to further explore ways to enhance LOT Merging's effectiveness on out-of-domain tasks in future work.

Next, we explored the use of domain-similar auxiliary data by applying a Gaussian blur to the original task exemplars. As shown in Table 12, LOT Merging consistently outperformed Task Arithmetic across all blur levels, although performance gradually declined as blur strength increased.

To investigate whether similar distribution exemplars would benefit, we merged BLIP models fine-tuned on COCO Caption, OKVQA, and ScienceQA, but replaced each task's exemplars with samples from semantically related datasets: Flickr30k for COCO Caption, AOKVQA for OKVQA, and IconQA for ScienceQA. As shown in Table 13, LOT Merging with surrogate exemplars outperforms Task Arithmetic, though the improvements are task-dependent—substantial for COCO Caption, but more limited for the other two.

### D.9 Robustness Analysis with Data Corruption

To assess the robustness of LOT Merging under noisy conditions, we adopted the corruption protocol introduced in [41], applying seven types of corruption to both exemplars and test sets. We evaluated the performance by merging eight ViT-B/32 models on each corrupted dataset and compared LOT Merging to Task Arithmetic. Table 14 shows that LOT Merging maintains consistent improvement over Task Arithmetic under all corruption types.

Table 14: Average accuracy under different corruption types

| Corruption Type | Task Arithmetic | LOT Merging ($\uparrow \Delta$) |
|---|---|---|
| Clean (no corruption) | 69.1 | 82.7 ($\uparrow$ 13.6) |
| Motion Blur | 47.3 | 59.9 ($\uparrow$ 12.6) |
| Impulse Noise | 51.1 | 60.8 ($\uparrow$ 9.7) |
| Gaussian Noise | 50.6 | 60.1 ($\uparrow$ 9.5) |
| Pixelate | 48.9 | 61.8 ($\uparrow$ 12.9) |
| Spatter | 65.0 | 79.2 ($\uparrow$ 14.2) |
| Contrast | 38.5 | 45.9 ($\uparrow$ 7.4) |
| JPEG Compression | 25.9 | 28.1 ($\uparrow$ 2.2) |

### D.10 Results on Merging Language Models

We conducted experiments using RoBERTa [16] as the backbone on the GLUE benchmark [33], which comprises eight diverse NLP tasks spanning both classification and regression (e.g., stsb). For evaluation, we report accuracy for classification tasks and the average of Pearson and Spearman correlations for the regression task. The results are presented in Table 15. LOT Merging achieves the highest average performance compared to the baseline methods, demonstrating its effectiveness for language model merging.

Table 15: Performance of Different Algorithms on GLUE Tasks

| Algorithm | CoLA | MNLI | MRPC | QNLI | QQP | RTE | SST-2 | STS-B | Average | #best |
|---|---|---|---|---|---|---|---|---|---|---|
| Task Arithmetic | 6.68 | 66.23 | 78.46 | 78.62 | 72.69 | 53.43 | 83.49 | 27.10 | 58.34 | 1 |
| Ties-Merging | 9.46 | 59.34 | 74.71 | 65.93 | 41.29 | 47.29 | 72.13 | 9.21 | 47.42 | 0 |
| PCB Merging | 11.40 | 50.85 | 77.63 | 78.22 | 55.78 | 60.29 | 75.57 | 67.01 | 59.59 | 1 |
| CAT Merging | 33.20 | 72.33 | 68.22 | 82.92 | 76.05 | 62.82 | 89.33 | 15.57 | 62.56 | 2 |
| LOT Merging | 17.13 | 73.04 | 78.27 | 77.22 | 78.81 | 65.35 | 89.74 | 25.50 | 63.13 | 4 |

Table 16: Performance Comparison when merging two Qwen models.

| Method | ArguAna | ArXiv Hierarchical Clustering S2S |
|---|---|---|
| Individual | 53.62 | 0.6377 |
| Task Arithmetic | 30.86 | 0.6318 |
| LOT Merging | 47.36 | 0.6374 |

We also conduct experiments on large language models. We select Qwen3-0.6B [39] as the pretrained backbone, and two official fine-tuned variants—Qwen3-Embedding-0.6B and Qwen3-Reranker-0.6B. Although Qwen3-0.6B has a relatively small parameter count, it features a deep architecture with 28 Qwen3DecoderBlocks, each comprising an attention module (four projection linear layers) and an MLP module (three projection linear layers), for a total of 196 linear layers.

To reflect their respective strengths in embedding and ranking, we selected two tasks from the MTEB (English, v2) benchmark [21]: ArguAna — a retrieval task requiring strong ranking capabilities; ArXiv Hierarchical Clustering S2S — a clustering task requiring high-quality embeddings. Evaluation metrics were top-1 accuracy for ArguAna and V-measure for ArXiv Hierarchical Clustering. Results in Table 16 demonstrate that LOT Merging substantially outperforms Task Arithmetic on ArguAna, indicating improved cross-task transfer for ranking, while maintaining competitive clustering performance. These results confirm that our method remains effective on deep LLMs such as Qwen3.

Table 17: Pearson Correlation across Different Tasks

| | COCO Caption | Flickr30k Caption | TextCaps | OKVQA | TextVQA | ScienceQA |
|---|---|---|---|---|---|---|
| Pearson Correlation | 0.51 | 0.60 | 0.30 | 0.82 | 0.77 | 0.20 |

## D.11 Analysis of Feature Drift on Multi-Model Tasks

To assess the impact of feature drift in vision-language settings, we measured the Pearson correlation between feature drift and loss change in BLIP models across several multimodal benchmarks after Task Arithmetic merging. The results in 17 demonstrate that feature drift remains a positively correlated factor.

While the correlation strength varies—reflecting inherent complexities such as reasoning demands and modality interactions—our experiments consistently show that reducing feature drift leads to performance gains, even in vision-language tasks. This suggests that, despite the greater complexity in multimodal models, minimizing feature drift remains an effective and broadly applicable merging objective.

## D.12 Robustness Analysis under Real-Time Data Distribution

We further study the impact of real-time data on model performance, particularly in scenarios where exemplars are available for only a subset of classes, while evaluation is conducted on the complete test set. The results in Table 18 show that although LOT Merging's performance declines with fewer exemplar classes, it still consistently outperforms Task Arithmetic.

Table 18: Effect of Exemplar Classes on Average Accuracy

| Backbone | Method | Exemplar Classes | Avg. Acc. |
|----------|--------|------------------|-----------|
| ViT-B/32 | Task Arithmetic | – | 69.1 |
| ViT-B/32 | LOT Merging | All classes | 82.7 |
| ViT-B/32 | LOT Merging | First 1/2 | 81.8 |
| ViT-B/32 | LOT Merging | First 1/3 | 81.0 |
| ViT-B/32 | LOT Merging | First 1/4 | 79.6 |
| ViT-B/32 | LOT Merging | First 1/5 | 77.6 |
| ViT-L/14 | Task Arithmetic | – | 84.5 |
| ViT-L/14 | LOT Merging | All classes | 90.5 |
| ViT-L/14 | LOT Merging | First 1/2 | 90.0 |
| ViT-L/14 | LOT Merging | First 1/3 | 89.6 |
| ViT-L/14 | LOT Merging | First 1/4 | 88.9 |
| ViT-L/14 | LOT Merging | First 1/5 | 88.3 |

