# OpenReview forum: "Towards Minimizing Feature Drift in Model Merging: Layer-wise Task Vector Fusion for Adaptive Knowledge Integration"
_NeurIPS.cc/2025/Conference — NeurIPS 2025 poster_

### Official Review · Reviewer_VNUV · 2025-06-29

**Clarity:** 4
**Significance:** 2
**Originality:** 3
**Rating:** 5
**Confidence:** 4

**Summary:**

The paper studies the problem of negative transfer in model merging. The authors argue that the degradation of merged model compared with individual model partly comes from feature drift. To solve this problem, they propose LOT merging, which minimizes such drift in a per-layer manner. The solution improves performance compared with baselines.

**Questions:**

1. [1] finds that when the original training set is unavailable, surrogate data from similar or related datasets can be useful for their localization step. I wonder whether it would also be the case for feature alignment? Would using similar data from other distribution be helpful for LOT merging? If so, it would widen the applicability of the approach.

[1] Localize-and-Stitch: Efficient Model Merging via Sparse Task Arithmetic. He et al, 2025.

**Ethical Concerns:**

["NO or VERY MINOR ethics concerns only"]

**Final Justification:**

The proposed approach is effective and technically sound. The rebuttal address the concerns, and the inclusion of surrogate data analysis extends the applicability of the approach.

**Limitations:**

Yes

**Quality:**

3

**Strengths And Weaknesses:**

Strength
1. The paper is well written, with claims supported by empirical and theoretical evidence.
2. The comparison with RegMean is clear and highlights the difference, strengthening the motivation.
3. The ablation studies clearly demonstrate the effect of different parameters.

Weakness
1. Similar to RegMean, the closed-form solution only applies to linear cases. Although such computation can be applied to each of the computation module separately, it is unclear why the aggregation would work for general transformer models, which are highly non-linear. It would be better to provide more intuition and explanation about this restriction.

---

> ### Author Rebuttal · Authors · 2025-07-31
>
> ### Response to **VNUV**
>
> We sincerely thank you for your critical assessment and constructive suggestions.
>
> **Q: How does the linear feature drift minimization affect the merging of nonlinear models?**
>
> A: Thank you for your thoughtful question. You are correct that the classical closed-form solution is strictly applicable to linear systems. However, our approach leverages the modular structure of modern deep neural networks, which are composed of sequences of linear modules interleaved with nonlinearities. While the overall model is highly nonlinear, we show that it is still possible to **bound the output drift of nonlinear modules** in terms of the drift in their underlying linear components.
>
> To illustrate this, consider the attention module, a central building block in transformer models. Its computation can be written as:
>
> $Att(Q, K, V) = \mathrm{Softmax}\left(\frac{QK^\top}{\sqrt{d}}\right) V,$
> where $d$ is the feature dimension. We establish the following proposition (with a proof sketch provided below):
>
> ---
>
> **Proposition 1.**
>
> Let $\Delta Q, \Delta K, \Delta V$ denote arbitrary perturbations to the input matrices. Then:
>
> $\|Att(Q, K, V) - Att(Q +\Delta Q, K +\Delta K, V + \Delta V)\|_F $
>
> $\leq \Delta_{linear} + \Delta_{non-linear}$
>
>
> where $\|\cdot\|_F$ denotes Frobenius norm and
>
> $$
> \Delta_{linear}=\frac{\|V\|_F \|Q\|_F}{\sqrt{d}} \|\Delta K\|_F+ \frac{\|V\|_F \|K\|_F}{\sqrt{d}} \|\Delta Q\|_F + \|\Delta V\|_F
> $$
>
> $$
> \Delta_{non-linear}=\frac{\|V\|_F}{\sqrt{d}} \|\Delta Q\|_F \|\Delta K\|_F
> $$
>
> ---
>
> **Proposition 1** shows that the total drift in the attention output—caused by perturbations to $Q$, $K$, and $V$—is bounded by a **sum of linear and nonlinear terms**. Crucially, when the perturbations $\Delta Q, \Delta K, \Delta V$ are small (which is precisely the regime targeted by LOT merging), the nonlinear term becomes negligible, and the bound is dominated by the linear terms. Furthermore, the coefficients of the linear terms are either constant or normalized by $\sqrt{d}$, ensuring that the overall drift remains controllable.
>
> This bound provides a theoretical justification for our decomposition approach: **by controlling the drift in the internal linear layers ($\Delta Q, \Delta K, \Delta V$), we can effectively bound the output drift of the entire attention module—even though the module itself is nonlinear**. This insight extends to other nonlinear modules in deep networks, provided we can analyze their dependence on internal linear computations.
>
> ---
>
> **Proof Sketch of Proposition 1:**
>
> Let
>
> $A = \frac{QK^\top}{\sqrt{d}}$
>
> $A' = \frac{(Q + \Delta Q)(K + \Delta K)^\top}{\sqrt{d}}$
>
> $\Delta A = A' - A$
>
> By the triangle inequality,
>
> $\|Att(Q, K, V) - Att(Q + \Delta Q, K + \Delta K, V + \Delta V)\|_F \leq \|\mathrm{Softmax}(A)V - \mathrm{Softmax}(A')V\|_F+\|\mathrm{Softmax}(A')\Delta V\|_F .$
>
> Since each row of $\mathrm{Softmax}(A')$ is a probability vector:
>
> $\|\mathrm{Softmax}(A')\Delta V\|_F \leq \|\Delta V\|_F$.
>
> By the Lipschitz continuity of softmax [1] and sub-multiplicativity of norms:
>
> $\|\mathrm{Softmax}(A)V - \mathrm{Softmax}(A')V\|_F \leq \|\mathrm{Softmax}(A) - \mathrm{Softmax}(A')\|_F \|V\|_F \leq \|\Delta A\|_F \|V\|_F.$
>
> Next,
>
> $\Delta A = \frac{Q \Delta K^\top + \Delta Q K^\top + \Delta Q \Delta K^\top}{\sqrt{d}}$
>
> so
>
> $\|\Delta A\|_F \leq \frac{\|Q\|_F \|\Delta K\|_F+\|K\|_F \|\Delta Q\|_F+\|\Delta Q\|_F \|\Delta K\|_F
> }{\sqrt{d}}$
>
> Substituting back yields the stated bound.
>
> ---
>
> **Reference:**
>
> [1] Gao, B., & Pavel, L. (2017). On the properties of the softmax function with application in game theory and reinforcement learning. arXiv preprint arXiv:1704.00805.
>
> **Q: Would using similar data from other distributions be helpful for LOT merging?**
>
> A: Thank you for raising this point.
>
> To investigate this, we merged BLIP models fine-tuned on COCO Caption, OKVQA, and ScienceQA, but replaced each task’s exemplars with samples from semantically related datasets: Flickr30k for COCO Caption, AOKVQA for OKVQA, and IconQA for ScienceQA. As shown below, LOT Merging with surrogate exemplars outperforms Task Arithmetic, though the improvements are task-dependent—substantial for COCO Caption, but more limited for the other two:
>
> |  | COCO Caption | OKVQA | ScienceQA |
> | --- | --- | --- | --- |
> | Task Arithmetic | 0.53 | 40.37 | 51.55 |
> | LOT Merging | 0.87 | 44.25 | 60.08 |
> | LOT Merging w/ surrogate exemplars | 0.62 | 40.15 | 53.54 |
>
> We hypothesize that this variability is due to differences in semantic similarity between the surrogate and original exemplars. To systematically test the effect of degraded similarity, we conducted additional experiments using the 8-task vision model merging setup, applying progressively stronger Gaussian blur to the exemplars.
>
> We observe that LOT Merging continues to outperform Task Arithmetic even as the exemplars are heavily degraded, though accuracy decreases as blur increases:
>
> |  | Gaussian Blur Kernel | Avg. Acc. when Merging ViT-B/32 (%) | Avg. Acc. when Merging ViT-L/14(%) |
> | --- | --- | --- | --- |
> | Task Arithmetic | - | 69.1 | 84.5 |
> | LOT Merging | $kernelsize=3, \sigma=1.0$ | 82.2 | 90.3 |
> | LOT Merging | $kernelsize=5, \sigma=2.0$ | 80.0 | 89.5 |
> | LOT Merging | $kernelsize=7, \sigma=3.0$ | 78.3 | 88.7 |
>
> Together, these findings suggest that **LOT Merging can indeed benefit from surrogate data, but the effectiveness depends on the semantic similarity to the original task**. This insight extends the application of LOT Merging when original data is unavailable.

---

> > ### Comment · Reviewer_VNUV · 2025-08-04
> >
> > Thank you for the detailed response and additional experiments. Those address my concerns, and I have raised my score.

---

> > > ### Author Response · Authors · 2025-08-06
> > >
> > > Thank you for your feedback. We're pleased to hear that our response has addressed your concerns, and we greatly appreciate your positive score. We would also like to extend our gratitude to the reviewer for the insightful suggestion regarding the exemplars from similar distributions, which has strengthened the paper.

---

### Official Review · Reviewer_NbB2 · 2025-06-29

**Clarity:** 2
**Significance:** 3
**Originality:** 2
**Rating:** 5
**Confidence:** 5

**Summary:**

The paper presents LOT Merging, a training-free model merging approach to tackle feature drift in multi-task scenarios. It reveals that performance degradation during merging correlates with feature drift, which is amplified in deeper network layers. LOT Merging mitigates this via layer-wise optimization, formulating model merging as a layer-wise feature drift minimization problem to derive closed-form solutions. It achieves up to 4.4% accuracy gains on vision tasks (leading in 6–7 out of 8 datasets for ViT models) and excels in 5 out of 6 vision-language tasks among the selected baselines.

**Questions:**

- Where the exemplar sets are selected from? Are they sampled from the training, validation, or test sets?
- Since the method optimizes an upper bound and decomposes the problem into layer-wise objectives, how does merging effectiveness scale with the number of layers? For instance, does LOT merging exhibit improved or degraded performance when applied to models with deeper architectures (e.g., 96-layer LLMs like Qwen)?

**Ethical Concerns:**

["NO or VERY MINOR ethics concerns only"]

**Final Justification:**

The addtional experiments solve my concerns about the differences with CAT Merging, the robustness of exemplars.
The effectiveness on LLMs and the theoretical treatment of nonlinear operations (like attention/SwiGLU) still can be improved.

**Limitations:**

The method relies on optimizing an upper bound, reducing the problem to layer-wise optimization. It depends critically on two key assumptions: the Lipschitz continuity and linear operations within different neural network layers and requires exemplars to estimate feature drift, which may pose practical challenge with modern architectures and noisy datasets. The applicability on LLMs and NLP tasks are also unverified.

**Paper Formatting Concerns:**

The spacing between Line 599 and Table 10 is too small

**Quality:**

3

**Strengths And Weaknesses:**

### Strengths
- The paper provides an algorithm that can work with few exemplars (16–64 samples per task) to mitigate the merging conflicts.
- The paper provides detail theoretical derivation in view of layer feature drifts.
- The performance of the proposed method is strong across vision-related benchmarks.
- The comparion of direct parameter merging and task vector merging, along with their dependency on samples size in Section 5.2, is interesing.

### Weaknesses
- The paper primarily employs the theoretical framework from Cat Merging [1] but does not explicitly discuss the relationship between the two methods, which may affect its originality. I am also interested in an analysis of why Cat Merging underperforms compared to LOT Merging, given that their primary difference lies in the optimization objective.
- The method performance may highly related to the quality of exemplars. While the authors report average performance across 3 randomly sampled exemplar sets, including the standard deviation would better illustrate consistency.
Furthermore, I think futher analysis of the robustness of LOT Merging against distributional differences is valuable. How does the method perform when exemplars/test sets are noisy (e.g.,  corruption data in Ada-Merging Figure 5 [1]),  or tasks involve complex patterns (e.g., NLP datasets with different instructions[2] ). In situations where access to exemplar sets is limited, or when real-time test data diverges from the distribution of exemplars, the applicability of LOT Merging may be constrained.
- Limited evaluation for larger models and NLP tasks. The experiments focus on vision tasks (ViT, BLIP), but omit testing on LLMs (e.g., Qwen, LLaMA) or complex benchmarks (e.g., GSM8K, MMLU). For large language models, does the $O(n^3)$ inverse operation efficient with high-dimensional layers (e.g., Qwen’s 12,288-dim FFN)? Does the layer-wise optimization perform well with deep architectures? Furthermore, The paper does not address attention mechanisms or SwiGLU activations, which are prevalent in modern LLMs.
- Derivation Errors:  In Equation 22-23 of Section B, should $f_k^{-1}$ instead be $f_k^{l-1}$ ?

[1] CAT Merging: A Training-Free Approach for Resolving Conflicts in Model Merging. ICML, 2025

[2] Adamerging: Adaptive model merging for multi-task learning. ICLR,2024

[3] Wizardlm: Empowering large pre-trained language models to follow complex instructions. ICLR,2024

---

> ### Author Rebuttal · Authors · 2025-07-31
>
> ### Response to **NbB2**
>
> We sincerely thank you for your critical assessment and constructive suggestions.
>
> **Q: Comparison between CAT Merging and LOT Merging.**
>
> A: Thanks for highlighting this point. While both CAT Merging and LOT Merging address layer-wise conflicts, they differ fundamentally in their treatment of task vectors:
>
> - **CAT Merging** uses **trimming** (projection and masking) to remove “conflict-prone” components. While this suppresses harmful interactions, it may also discard critical task-specific directions, particularly when those directions contain information valuable for other tasks.
> - **LOT Merging**, by contrast, **directly minimizes layer-wise feature drift** via a convex quadratic formulation. This closed-form solution preserves all informative subspaces and optimally balances contributions from each task. In Section 5.1, we show that in both the ideal (orthogonal features) and worst (collinear features) cases, LOT Merging achieves the optimal consolidation of task vectors—something trimming cannot guarantee.
>
> Empirically, CAT’s trimming sometimes over-prunes useful information, whereas LOT Merging shows more consistent improvements (in Table 10, Section D.7). We will clarify these distinctions in the revised manuscript.
>
> **Q: The robustness of LOT Merging under corruption exemplars/test sets.**
>
> A: We appreciate the reviewer’s insightful comments regarding the robustness of LOT Merging.
>
> To assess the robustness of LOT Merging under noisy conditions, we adopted the corruption protocol introduced in Ada-Merging [1], applying seven types of corruption to both exemplars and test sets. We evaluated the performance by merging **eight** ViT-B/32 models on each corrupted dataset and compared LOT Merging to Task Arithmetic. The following results show that LOT Merging maintains consistent improvement over Task Arithmetic under all corruption types.
>
> | Corruption Type | Task Arithmetic | LOT Merging (↑ Δ) |
> | --- | --- | --- |
> | Clean (no corruption) | 69.1 | **82.7 (↑ 13.6)** |
> | Motion Blur | 47.3 | **59.9 (↑ 12.6)** |
> | Impulse Noise | 51.1 | **60.8 (↑ 9.7)** |
> | Gaussian Noise | 50.6 | **60.1 (↑ 9.5)** |
> | Pixelate | 48.9 | **61.8 (↑ 12.9)** |
> | Spatter | 65.0 | **79.2 (↑ 14.2)** |
> | Contrast | 38.5 | **45.9 (↑ 7.4)** |
> | JPEG Compression | 25.9 | **28.1 (↑ 2.2)** |
>
> **Q: Results on NLP tasks & the robustness analysis of LOT Merging with different instructions.**
>
> A: Thanks for your suggestion. We follow CAT Merging and conduct experiments using RoBERTa as the backbone model on the GLUE benchmark, including eight diverse NLP tasks spanning both classification and regression (e.g., STS-B). For evaluation, we report accuracy for classification tasks and the average of Pearson and Spearman correlations for the regression task. The results are summarized in the table below. Our proposed LOT Merging achieves the highest average performance compared to baseline methods.
>
> Notably, we observe that LOT Merging exhibits superior robustness on tasks with more complex or nuanced instructions, such as:
>
> - rte demands precise logical inference between sentence pairs.
> - qqp involves paraphrase detection, relying on intent and semantic understanding rather than surface patterns.
> - sst2 requires fine-grained sentiment classification, sensitive to subtle linguistic cues.
>
> In contrast, some baseline methods tend to overfit tasks with simpler instructions, such as stsb, leading to less balanced performance. These results suggest that LOT Merging is more robust across tasks with varying levels of instruction complexity.
>
> | **Algorithm** | **cola** | **mnli** | **mrpc** | **qnli** | **qqp** | **rte** | **sst2** | **stsb** | **Average** | #best |
> | --- | --- | --- | --- | --- | --- | --- | --- | --- | --- | --- |
> | Task Arithmetic | 6.68 | 66.23 | **78.46** | 78.62 | 72.69 | 53.43 | 83.49 | 27.10 | 58.34 | 1 |
> | Ties-Merging | 9.46 | 59.34 | 74.71 | 65.93 | 41.29 | 47.29 | 72.13 | 9.210 | 47.42 | 0 |
> | PCB Merging | 11.40 | 50.85 | 77.63 | 78.22 | 55.78 | 60.29 | 75.57 | **67.01** | 59.59 | 1 |
> | CAT Merging | **33.20** | 72.33 | 68.22 | **82.92** | 76.05 | 62.82 | 89.33 | 15.57 | 62.56 | 2 |
> | LOT Merging | 17.13 | **73.04** | 78.27 | 77.22 | **78.81** | **65.35** | **89.74** | 25.5 | **63.13** | 4 |
>
> **Q: The robustness of LOT Merging under real-time data distribution.**
>
> We further study the impact of real-time data on model performance, particularly in scenarios where exemplars are available for only a subset of classes, while evaluation is conducted on the complete test set. The results below show that although LOT Merging’s performance declines with fewer exemplar classes, it still consistently outperforms Task Arithmetic.
>
> | Backbone | Method | Exemplar Classes | Avg. Acc. |
> | --- | --- | --- | --- |
> | ViT-B/32 | Task Arithmetic | - | 69.1 |
> |  | LOT Merging | All classes | 82.7 |
> |  | LOT Merging | First 1/2 classes | 81.8 |
> |  | LOT Merging | First 1/3 classes | 81.0 |
> |  | LOT Merging | First 1/4 classes | 79.6 |
> |  | LOT Merging | First 1/5 classes | 77.6 |
> |  |  |  |  |
> | ViT-L/14 | Task Arithmetic | - | 84.5 |
> |  | LOT Merging | All classes | 90.5 |
> |  | LOT Merging | First 1/2 classes | 90.0 |
> |  | LOT Merging | First 1/3 classes | 89.6 |
> |  | LOT Merging | First 1/4 classes | 88.9 |
> |  | LOT Merging | First 1/5 classes | 88.3 |
>
> **Q: Effectiveness for larger models.**
>
> A: Good question. To assess computational feasibility, we empirically measured the time required to compute the Moore–Penrose inverse of a randomly initialized 12,288×12,288 matrix across several platforms:
>
> | Device | Xeon Gold (cpu) | Xeon Platinum (cpu) | RTX3090 (gpu) | A100 (gpu) |
> | --- | --- | --- | --- | --- |
> | Time (second) | 87.20 | 36.51 | 79.02 | 22.91 |
>
> These results confirm that the inverse operation is tractable in practice. Even on the Xeon Gold, performing such inversion across 96 layers requires approximately 139.5 minutes. Given the one-time cost and the absence of any retraining, we consider this overhead acceptable at LLM scale. Furthermore, our method only invokes the pseudo-inverse on exemplar-aggregated covariance matrices, which are typically low-rank in practice, further reducing computational burden.
>
> In addition, following [1], we conducted preliminary experiments (due to time constraints) on merging two LLaMA-2-13B-based models: WizardLM-13B, fine-tuned for instruction following (evaluated on AlpacaEval 2.0), and WizardMath-13B, specialized for mathematical reasoning (evaluated on the MATH dataset). As shown in the table below, LOT Merging outperforms Task Arithmetic on both benchmarks, although the margin is relatively modest.
>
> |  | AlpacaEval 2.0 | MATH | Average |
> | --- | --- | --- | --- |
> | Task Arithmetic | 10.91 | 12.21 | 11.56 |
> | LOT Merging | 11.02 | 12.40 | 11.71 |
>
> [1] Yu L, Yu B, Yu H, et al. Extend model merging from fine-tuned to pre-trained large language models via weight disentanglement[J]. arXiv preprint arXiv:2408.03092, 2024.
>
> **Q: How LOT Merging tackle attention mechanisms or SwiGLU activations.**
>
> A: Thank you for your question. Our approach decomposes complex modules into their fundamental parameterized linear operations:
>
> - **Attention Mechanism:** The Q, K, V projections and the output projection are each treated as distinct linear layers and merged independently. Our experiments with attention-based models such as ViT and BLIP demonstrate the effectiveness of this decomposition.
> - **SwiGLU Activation:** The gate and linear projections in SwiGLU are treated as two separate matrix multiplications, which can be merged independently as well.
>
> This modular treatment allows LOT Merging to naturally extend to attention mechanisms and SwiGLU activations, without requiring any additional modifications.
>
> **Q: Where are the exemplar sets selected from?**
>
> A: All exemplar sets are randomly sampled from the training set and are kept strictly separate from the test set used for evaluation.
>
> **Q: Writing issues (standard deviation and equation error in Section B).**
>
> A: Thank you for highlighting these issues. We will thoroughly revise the manuscript, correct the notation, and include the standard deviation in the experiments in the final version.

---

> > ### Comment · Reviewer_NbB2 · 2025-08-04
> >
> > Thanks for the authors' detail responses. The addtional experiments solve most of my concerns and would be a valuable addition to the final version of the paper. I have raised the scores but I still have some suggestions:
> >
> > - For the effectiveness analysis, reporting computation time across models of varying sizes rather than on different hardware platforms would provide a more convincing evaluation. The reported 139.5 minutes seems relatively slow; the authors may consider exploring additional speedup techniques.
> > - LLaMA 13B may be old, newer model like Qwen3 should be added.
> > - For the  attention or SwiGLU, we indeed can handle each linear weight independently. However, as the theory is derived based on the linear operation assupution, this may not yield optimal performance in practice. For the more broader applicability, I suggest authors extend their theoretical analysis with explicit derivations tailored to attention and SwiGLU operations.

---

> > > ### Author Response · Authors · 2025-08-06
> > >
> > > Thank you for your feedback. We are glad to hear that our response has addressed most of your concerns, and we appreciate your increased score. Below, you'll find our responses to the remaining suggestions.
> > >
> > > **Q: The effectiveness of LOT merging.**
> > >
> > > A: Thanks for the suggestion. We acknowledge that the naive matrix inversion involved in computing the optimal merging vector $T^*=A^{-1}B$ ($A=\sum_kX_k^{\top}X_k$, $B=\sum_kX_k^{\top}X_kT_k$) does incur a high computational cost.
> > >
> > > To address this, we adopted a linear fitting-based approach by directly solving $AT^*=B$. which eliminates the need for explicit matrix inversion. This method is also efficiently implemented on GPUs using `torch.linalg.solve(A, B)`. Importantly, it maintains robust performance: in our experiments merging eight ViT-B/32 models, the linear fitting method achieved an average accuracy of 82.6, nearly matching the 82.7 obtained with the naive inversion.
> > >
> > > We also simulated the merging time on a single H100 GPU of both methods on the Qwen3 model family, with results summarized below. As shown, the linear fitting approach delivers **up to 22× speedup** on the largest model, making it highly practical for large-scale model merging.
> > >
> > > |  | **Inversion (minnutes)** | **Linear Fitting (minnutes)** |
> > > | --- | --- | --- |
> > > | Qwen3-4B | 10.73 | 1.09 |
> > > | Qwen3-8B | 19.91 | 2.63 |
> > > | Qwen3-14B | 57.80 | 3.88 |
> > > | Qwen3-32B | 264.49 | 12.09 |
> > >
> > > **Q: LLaMA 13B may be old, a newer model like Qwen3 should be added.**
> > >
> > > A: Thank you for the valuable suggestion. We agree that incorporating a newer model like Qwen3 would strengthen our evaluation. While Qwen3 is a very recent release and suitable finetuned checkpoints are still limited, we are actively working to include related experiments. We aim to provide more comprehensive comparisons with Qwen3 in future revisions.
> > >
> > > **Q: Extend theoretical analysis with explicit derivations tailored to attention and SwiGLU operations.**
> > >
> > > A: We appreciate the reviewer’s insightful suggestion. Our current theoretical analysis is indeed based on the assumption of purely linear operations, whereas both multi-head attention and SwiGLU introduce complex nonlinearities (i.e., softmax and Swish gating), which render the resulting adaptation objectives non-convex. At present, we are not aware of any efficient closed-form solutions for these more complex optimization problems. In practice, one common approach is to apply a first-order Taylor expansion, which linearizes the objective and reduces it to a least-squares problem, as done in our LOTMerging framework. Alternatively, more general iterative numerical methods—such as ADMM or BFGS—can be used, though these typically incur much higher computational costs.
> > >
> > > We agree that developing efficient, specialized solvers for attention mechanisms and SwiGLU would substantially broaden the applicability of our approach. We thank the reviewer for highlighting this important direction, and we consider it a promising avenue for future research.

---

> > > ### Author Response · Authors · 2025-08-08
> > >
> > > **Q: Performance of LOT Merging on Qwen3.**
> > >
> > > A: We have now finished a preliminary evaluation on Qwen3. Specifically, we select Qwen3-0.6B as the pretrained backbone, and two official fine-tuned variants—Qwen3-Embedding-0.6B and Qwen3-Reranker-0.6B. Although Qwen3-0.6B has a relatively small parameter count, it features a deep architecture with 28 Qwen3DecoderBlocks, each comprising an attention module (four projection linear layers) and an MLP module (three projection linear layers), for a total of 196 linear layers.
> > >
> > > To reflect their respective strengths in embedding and ranking, we selected two tasks from the MTEB (English, v2) benchmark:
> > >
> > > - ArguAna — a retrieval task requiring strong *ranking* capabilities.
> > > - ArXiv Hierarchical Clustering S2S — a clustering task requiring high-quality *embeddings*.
> > >
> > > Evaluation metrics were top-1 accuracy for ArguAna and V-measure for ArXiv Hierarchical Clustering. Results are as follows:
> > >
> > > |  | ArguAna | ArXiv Hierarchical Clustering S2S |
> > > | --- | --- | --- |
> > > | Individual | 53.62 | 0.6377 |
> > > | Task Arithmetic | 30.86 | 0.6318 |
> > > | LOT Merging | 47.36 | 0.6374 |
> > >
> > > LOT Merging substantially outperforms Task Arithmetic on ArguAna, indicating improved cross-task transfer for ranking, while maintaining competitive clustering performance. These results confirm that our method remains effective on deep LLMs such as Qwen3.
> > >
> > > We sincerely thank the reviewer for the valuable suggestions. They have greatly enhanced the quality of our work.

---

### Official Review · Reviewer_eNpX · 2025-07-02

**Clarity:** 3
**Significance:** 3
**Originality:** 3
**Rating:** 5
**Confidence:** 3

**Summary:**

This paper introduces Layer-wise Optimal Task Vector Merging (LOT Merging) , a training-free approach to model merging that minimizes feature drift between task-specific expert models and the resulting merged model, operating layer by layer. In the context of Transformer-based models, operations are categorized into three types: matrix multiplication, element-wise product, and element-wise addition. And tailored merging strategies for each type of operation are proposed. LOT Merging is evaluated across a range of vision tasks as well as vision-language tasks.

**Questions:**

Please refer to Weaknesses. Overall I think this is a nice paper, I have some concerns as raised in Weaknesses. My initial rating is BA.

**Ethical Concerns:**

["NO or VERY MINOR ethics concerns only"]

**Final Justification:**

The rebuttal addressed my concerns. Having read other reviews, I've decided to keep my original positive assessment of this work, and increase the score to 5.

**Limitations:**

yes

**Quality:**

3

**Strengths And Weaknesses:**

Strengths:
* The paper is well-structured and clearly written, making it easy to follow.
* The core idea is intuitive and supported by theoretical analysis.
* The method is validated across a variety of tasks, accompanied by thorough ablation studies that provide a comprehensive understanding of its behaviour.

Weaknesses:
* LOT Merging is based on the assumption that feature drift has a significant impact on the performance of merged models. This seems reasonable for relatively simple unimodal tasks, such as those in vision-only settings demonstrated in Task Arithmetic (e.g., Figure 1). However, I question whether this relationship remains as clear-cut for more complex tasks, such as vision-language tasks or image generation, where interactions between modalities may introduce additional challenges and obscure straightforward patterns of feature drift.
* Related to task complexity, LOT Merging relies on examplars to estimate feature drift. When dealing with highly complex tasks, the limited number of exemplars may fail to accurately represent the overall data distribution, potentially leading to unstable performance, this is somewhat reflected in the results on OKVQA in Table 3. Additionally, it would be helpful to clarify what the Individual results refer to in that table.

---

> ### Author Rebuttal · Authors · 2025-07-31
>
> ### Response to **eNpX**
>
> We sincerely thank you for your critical assessment and constructive suggestions.
>
> **Q: Does feature drift still have an impact on multimodal tasks?**
>
> A: Thank you for raising this important point.
>
> To assess the impact of feature drift in vision-language settings, we measured the Pearson correlation between feature drift and loss change in BLIP models across several multimodal benchmarks after Task Arithmetic merging. The results below demonstrate that feature drift remains a positively correlated factor:
>
> |  | COCO Caption  | Flickr30k Caption | Textcaps | OKVQA | TextVQA | ScienceQA |
> | --- | --- | --- | --- | --- | --- | --- |
> | Pearson Correlation | 0.51 | 0.60 | 0.30 | 0.82 | 0.77 | 0.20 |
>
> While the correlation strength varies—reflecting inherent complexities such as reasoning demands and modality interactions—our experiments (see Table 3 in the manuscript) consistently show that reducing feature drift leads to performance gains, even in vision-language tasks. This suggests that, despite the greater complexity in multimodal models, minimizing feature drift remains an effective and broadly applicable merging objective.
>
> **Q: Can the limited number of exemplars lead to unstable performance?**
>
> A: Thank you for the thoughtful comments.
>
> To assess whether limited exemplars affect merging performance, we varied the number of exemplars per task from 1 to 64 and report LOT Merging accuracy on both vision and vision-language tasks. As shown below, increasing the number of exemplars generally improves performance, and most tasks reach near–peak performance by 32 samples. This indicates that the number of exemplars alone cannot fully explain the marginal gain on OKVQA in Table 3.
>
> | Num. of Exemplars | SUN397 | Cars | RESISC45 | EuroSAT | SVHN | GTSRB | MNIST | DTD |
> | --- | --- | --- | --- | --- | --- | --- | --- | --- |
> | 1 | 62.0 | 59.9 | 67.0 | 57.1 | 78.7 | 66.9 | 87.3 | 47.0 |
> | 2 | 63.2 | 60.5 | 74.2 | 71.4 | 87.5 | 72.9 | 95.3 | 47.5 |
> | 4 | 64.2 | 64.2 | 69.1 | 84.4 | 88.6 | 77.9 | 96.5 | 51.1 |
> | 8 | 65.4 | 66.3 | 79.4 | 87.6 | 92.8 | 85.5 | 98.2 | 54.1 |
> | 16 | 66.4 | 67.1 | 82.6 | 90.9 | 92.9 | 86.1 | 98.5 | 58.1 |
> | 32 | 67.4 | 67.5 | 83.3 | 92.4 | 93.2 | 89.6 | 98.6 | 60.6 |
> | 64 | 67.7 | 67.5 | 85.7 | 94.9 | 93.4 | 89.8 | 98.7 | 63.6 |
>
> | Num. of Exemplars | COCO Caption | Flickr30k Caption | Textcaps | OKVQA | TextVQA | ScienceQA |
> | --- | --- | --- | --- | --- | --- | --- |
> | Metric  | CIDEr | CIDEr | CIDEr | Accuracy | Accuracy | Accuracy |
> | 2 | 0.83 | 0.39 | 0.39 | 27.95 | 13.04 | 47.14 |
> | 4 | 0.83 | 0.46 | 0.41 | 31.86 | 14.70 | 47.68 |
> | 8 | 0.88 | 0.48 | 0.43 | 36.22 | 18.43 | 47.47 |
> | 16 | 0.89 | 0.53 | 0.45 | 37.34 | 20.12 | 47.37 |
> | 32 | 0.91 | 0.54 | 0.44 | 38.35 | 20.82 | 48.24 |
>
> Regarding the OKVQA results in Table 3, we would like to clarify that LOT Merging only underperforms CAT Merging on this particular dataset. Importantly, LOT Merging improves upon Task Arithmetic across *all* vision-language tasks, while CAT Merging degrades performance on Textcaps. We therefore attribute CAT’s large gain on OKVQA to task-specific overfitting, rather than to generally superior alignment. A similar observation also applies to our vision-only benchmarks (see Section D.7).
>
> **Q: Clarify the individual results of Table 3.**
>
> A: Thank you for the suggestion. We will include the individual results in the final version. For reference, the detailed results are as follows:
> |  | COCO Caption | Flickr30k Caption | Textcaps | OKVQA | TextVQA | ScienceQA |
> | --- | --- | --- | --- | --- | --- | --- |
> | Metric  | CIDEr | CIDEr | CIDEr | Accuracy | Accuracy | Accuracy |
> | **Individual** | 1.17 | 0.65 | 0.65 | 50.84 | 29.79 | 76.89 |
> | Task Arithmetic | 0.86 | 0.50 | 0.39 | 17.71 | 0.49 | 40.10 |
> | CAT Merging | 0.91 | 0.53 | 0.36 | 44.07 | 19.69 | 46.36 |
> | LOT Merging | 0.91 | 0.54 | 0.44 | 38.35 | 20.82 | 48.24 |

---

> > ### Comment · Reviewer_eNpX · 2025-08-05
> >
> > Thanks for the detailed response. I'm satisfied with the additional results. I'll keep my positive assessment of this work.

---

> > > ### Author Response · Authors · 2025-08-06
> > >
> > > Thank you for your feedback. We are pleased to hear that our response has addressed your concerns, and we greatly appreciate the positive score. We would also like to thank the reviewer for the insightful suggestion regarding the generalization of LOT merging in multi-model scenarios and the number of exemplars, which has further strengthened the paper.

---

### Official Review · Reviewer_ZqXr · 2025-07-03

**Clarity:** 3
**Significance:** 3
**Originality:** 2
**Rating:** 5
**Confidence:** 5

**Summary:**

The paper titled “Towards Minimizing Feature Drift in Model Merging: Layer-wise Task Vector Fusion for Adaptive Knowledge Integration” introduces a training-free approach called Layer-wise Optimal Task Vector Merging (LOT Merging) to mitigate feature drift in merged models. The method frames the merging process as a closed-form convex quadratic optimization problem, aiming to minimize representation differences at each layer. It does this at a task-vector level instead of parameter level avoiding the issue with modifying the pretrained weights which may lead to catastrophic forgetting.

**Questions:**

- While the method requires access to a small amount of task-specific data to generate the necessary activations, it would be interesting to explore whether task-agnostic auxiliary data could serve as an effective alternative.
- It remains unclear whether the performance gains observed on in-domain data would generalize to out-of-domain scenarios—an area where traditional merging methods typically excel. This concern is especially relevant since the proposed method focuses on minimizing feature drift specifically using in-domain data.

**Ethical Concerns:**

["NO or VERY MINOR ethics concerns only"]

**Final Justification:**

I'd like to thank the authors for their response. I'm happy with the additional evaluation to test generalisation and evlauations in the language domain and would like to maintain my positive rating.

**Limitations:**

yes

**Quality:**

3

**Strengths And Weaknesses:**

### Strengths
- The work proposes a novel technique to reduce layer-wise feature drift using a closed-form solution at the task-vector level, rather than the parameter level—a direction that has not been previously explored.
- It demonstrates strong performance, significantly closing the gap with jointly trained models, and benchmarks favorably against a wide range of training-free model merging baselines.

### Weaknesses
- The paper lacks an evaluation of the merged model's generalization performance—an aspect where merging often excels, sometimes outperforming individual task-specific models.
- It does not assess the method on large language models (LLMs), a domain where merging techniques are widely adopted. Such an evaluation could broaden its relevance to more popular and practical use cases.
- The method is described as “data-free,” though this may be misleading. It would be more accurate to position it as suited for data-scarce scenarios rather than completely data-free.

---

> ### Author Rebuttal · Authors · 2025-07-31
>
> ### Response to **ZqXr**
>
> We sincerely thank you for your critical assessment and constructive suggestions.
>
> **Q: Evaluation of the generalization performance.**
>
> Thank you for your thoughtful suggestion. To address concerns about generalization, we evaluated our method by merging models trained on six out of eight tasks and then testing on the remaining two unseen tasks.
>
> As shown in the tables below, LOT Merging consistently outperforms both Task Arithmetic and Ties-Merging—not only on seen tasks, but also on unseen (out-of-domain) tasks. These results demonstrate the moderate generalization ability of our approach. However, the performance gap does become narrower on out-of-domain tasks compared to in-domain tasks. This observation motivates us to further explore ways to enhance LOT Merging’s effectiveness on out-of-domain tasks in future work.
>
> |  | SUN397 |  Cars | RESISC45 | DTD | SVHN | GTSRB | **Avg Acc (Seen Tasks)** | MNIST (Unseen) | EuroSAT (Unseen) | **Avg Acc (Unseen Tasks)** |
> | --- | --- | --- | --- | --- | --- | --- | --- | --- | --- | --- |
> | Task Arithmetic | 63.3 | 62.4 | 75.1 | 57.8 | 84.6 | 80.4 | **70.6** | 77.2 | 46.2 | **61.7** |
> | Ties-Merging | 67.8 | 66.2 | 77.2 | 56.7 | 77.1 | 70.9 | **69.3** | 75.9 | 43.3 | **59.6** |
> | LOT Merging | 69.1 | 69.1 | 89.7 | 65.6 | 94.1 | 93.7 | **80.2** | 81.5 | 45.4 | **63.5** |
> |  | SUN397 |  Cars | GTSRB | EuroSAT | DTD | MNIST | **Avg Acc (Seen Tasks)** | RESISC45 (Unseen) | SVHN (Unseen) | **Avg Acc (Unseen Tasks)** |
> | Task Arithmetic | 64.0 | 64.0 | 75.2 | 87.7 | 57.0 | 95.7 | **73.9** | 52.3 | 44.9 | **51.1** |
> | Ties-Merging | 68.0 | 67.1 | 67.7 | 78.4 | 56.5 | 92.8 | **71.8** | 58.7 | 49.2 | **53.9** |
> | LOT Merging | 68.8 | 69.9 | 84.8 | 95.8 | 61.5 | 97.5 | **79.7** | 56.3 | 54.6 | **55.5** |
>
> **Q: Evaluation on large language models (LLMs).**
>
> A: Thank you for your suggestion. To address this point, we followed CAT Merging [1] and conducted experiments using RoBERTa as the backbone on the GLUE benchmark, which comprises eight diverse NLP tasks spanning both classification and regression (e.g., stsb). For evaluation, we report accuracy for classification tasks and the average of Pearson and Spearman correlations for the regression task. The results are presented in the table below. LOT Merging achieves the highest average performance compared to the baseline methods, demonstrating its effectiveness for language model merging.
>
> | **Algorithm** | **cola** | **mnli** | **mrpc** | **qnli** | **qqp** | **rte** | **sst2** | **stsb** | **Average** | #best |
> | --- | --- | --- | --- | --- | --- | --- | --- | --- | --- | --- |
> | Task Arithmetic | 6.68 | 66.23 | **78.46** | 78.62 | 72.69 | 53.43 | 83.49 | 27.10 | 58.34 | 1 |
> | Ties-Merging | 9.46 | 59.34 | 74.71 | 65.93 | 41.29 | 47.29 | 72.13 | 9.210 | 47.42 | 0 |
> | PCB Merging | 11.40 | 50.85 | 77.63 | 78.22 | 55.78 | 60.29 | 75.57 | **67.01** | 59.59 | 1 |
> | CAT Merging | **33.20** | 72.33 | 68.22 | **82.92** | 76.05 | 62.82 | 89.33 | 15.57 | 62.56 | 2 |
> | LOT Merging | 17.13 | **73.04** | 78.27 | 77.22 | **78.81** | **65.35** | **89.74** | 25.5 | **63.13** | 4 |
>
> In addition, following [2], we conducted preliminary experiments (due to time constraints) on merging two LLaMA-2-13B-based models: WizardLM-13B, fine-tuned for instruction following (evaluated on AlpacaEval 2.0), and WizardMath-13B, specialized for mathematical reasoning (evaluated on the MATH dataset). As shown in the table below, LOT Merging outperforms Task Arithmetic on both benchmarks, although the margin is relatively modest.
>
> |  | AlpacaEval 2.0 | MATH | Average |
> | --- | --- | --- | --- |
> | Task Arithmetic | 10.91 | 12.21 | 11.56 |
> | LOT Merging | 11.02 | 12.40 | 11.71 |
>
> [1] Sun W, Li Q, Geng Y, et al. CAT Merging: A Training-Free Approach for Resolving Conflicts in Model Merging[C]//ICML2025.
>
> [2] Yu L, Yu B, Yu H, et al. Extend model merging from fine-tuned to pre-trained large language models via weight disentanglement[J]. arXiv preprint arXiv:2408.03092, 2024.
>
> **Q: The description of “data-free” may be misleading.**
>
> A: Thank you for raising this point. We would like to clarify that we do not describe our method as “data-free” in the paper. Instead, we refer to it as “training-free,” indicating that it does not require any gradient-based optimization. We will make this distinction more explicit in the revised manuscript to avoid any potential confusion.
>
> **Q: Exploring the benefit of task-agnostic auxiliary data.**
>
> A: Thank you for this interesting suggestion.
>
> To investigate the potential of task-agnostic auxiliary data, we first experimented with using ImageNet samples as exemplars for all tasks. However, when merging eight ViT-B/32 models, we observed a noticeable performance drop compared to Task Arithmetic, as shown in the table below:
>
> |  | Task Arithmetic | LOT Merging w/ 64 ImageNet Exemplars | LOT Merging w/ 128 ImageNet Exemplars | LOT Merging w/ 256 ImageNet Exemplars |
> | --- | --- | --- | --- | --- |
> | Avg. Acc. (%) | 69.1 | 65.6 | 65.1 | 65.3 |
>
> Next, we explored the use of *domain-similar* auxiliary data by applying a Gaussian blur to the original task exemplars. In this setting, LOT Merging consistently outperformed Task Arithmetic across all blur levels, although performance gradually declined as blur strength increased:
>
> |  | Gaussian Blur Kernel | Avg. Acc. when Merging ViT-B/32 (%) | Avg. Acc. when Merging ViT-L/14(%) |
> | --- | --- | --- | --- |
> | Task Arithmetic | - | 69.1 | 84.5 |
> | LOT Merging | $kernelsize=3, \sigma=1.0$ | 82.2 | 90.3 |
> | LOT Merging | $kernelsize=5, \sigma=2.0$ | 80.0 | 89.5 |
> | LOT Merging | $kernelsize=7, \sigma=3.0$ | 78.3 | 88.7 |
>
> These results suggest that fully task-agnostic data can be detrimental to performance, whereas *semantically similar* data—even when degraded—can still facilitate effective merging. This insight could help extend LOT Merging to more practical or data-constrained scenarios.

---

> > ### Comment · Reviewer_ZqXr · 2025-08-06
> >
> > I'd like to thank the authors for their response. I'm happy with the additional evidence provided and would like to maintain my positive rating.

---

> > > ### Author Response · Authors · 2025-08-06
> > >
> > > Thank you for your feedback. We are glad that our response has addressed your concerns, and we greatly appreciate your support. We would also like to sincerely thank the reviewer for the insightful suggestions regarding experiments on LLMs, generalization, and task-agnostic auxiliary data, which have helped further strengthen the paper.

---

### Note · Authors · 2025-08-12

**Final Response**

We would like to express our sincere gratitude to the area chair and reviewers for their valuable suggestions and thoughtful effort in reviewing our submission. Your feedback has been instrumental in improving the quality of our work.

We greatly appreciate the reviewers’ recognition of the key contributions of our paper. We are also pleased that the reviewers found the thoroughness of our experiments and the clarity of our presentation to be strong aspects of the paper.

In response to the reviewers’ concerns raised in the rebuttal, we have made significant revisions to address the majority of the issues. We have clarified several aspects of the methodology and expanded the experimental results to further validate the effectiveness of the proposed method. We believe these revisions enhance the overall clarity and quality of the paper.

We assure that all of the changes discussed in the rebuttal will be incorporated into the final version of the paper. We believe these updates will improve the quality of our submission.

Once again, thank you for your thoughtful feedback.

Best regards,

Authors of Submission 15435

---

### Decision · Program_Chairs · 2025-09-17

**Decision:**

Accept (poster)

**Comment:**

The paper received mostly positive reviews initially. After the rebuttal, the reviewers' concerns have been addressed and agreed to accept the paper. The final decision of AC is consistent with the reviewers.